# Functionally deficient *UBOX5* variants and primary angle-closure glaucoma

Primary angle-closure glaucoma is a major cause of irreversible blindness worldwide afflicting >20 million people. Through whole exome sequencing, we analysed the association between gene-based burden of rare, protein-altering genetic variants and disease risk in 4,667 affected individuals and 5,473 unaffected controls. We tested genes surpassing exome-wide significance ($P < 2.5 \times 10^{-6}$) for replication in a further 2,519 cases and 472,189 controls. We observed carriers of rare, protein-altering variants at *UBOX5* (observed in 154 out of 7,186 affected individuals [2.1%] and in 3,975 out of 477,197 unaffected controls [0.83%]) to be associated with 2.13-fold increased risk of PACG (95%ci, 1.69 – 2.69; $P = 1.25 \times 10^{-10}$). We performed substrate trapping assays coupled with mass spectrometry and observed Binding Immunoglobulin Protein (BIP) as a key substrate for UBOX5. Biological assays showed UBOX5 acts by ubiquitinating BIP. We evaluated the functional status of 35 *UBOX5* variants and observed that functionally deficient variants were enriched in affected individuals compared to controls. We validated this finding in an independent collection where 3 persons carrying functionally deficient variants were observed out of 208 cases (1.4%), whereas none were observed in 600 controls. Our findings suggest the UBOX5−BIP signalling pathway might be involved in biology of primary angle-closure glaucoma.

Glaucoma causes irreversible blindness. Primary angle-closure glaucoma (PACG) is a major form of glaucoma that disproportionately affects persons of Asian ancestry, with >20 million people afflicted worldwide[1,2]. Characterized by obstruction of aqueous humour outflow in the anterior chamber angle of the eye, acutely or chronically elevated intraocular pressure (IOP), and vision loss due to optic nerve damage, PACG is >2.5 times as likely to cause blindness than other forms of glaucoma[2,3]. PACG often eludes detection and can present as an emergency with acute primary angle closure (APAC). To develop new therapeutic strategies against PACG, improved understanding of disease biology is needed.

First degree relatives of PACG patients have up to 3-fold increased disease risk compared to the general population[4], and reports of genetic mutations clustering in affected family members suggest a hereditable etiology[5]. While genome-wide association studies have identified significantly associated loci[6], pinpointing the exact causal genes within these loci is challenging due to the analysis of non-coding variants. This limitation hinders understanding the biological mechanisms of PACG[7]. On the other hand, whole exome sequencing enumerates variants in the protein-coding sequence, offering direct insights into disease biology.

We hypothesize that there are genetic variants with strong effect on PACG risk residing in the protein-coding regions (exome) of the genome. We sequenced the exomes of 4,667 patients with PACG and 5,473 unaffected individuals from Singapore, Hong Kong, Japan, and Vietnam. Taking each gene as the unit of analysis, all qualifying variants (defined as variants predicted by the CADD algorithm[7] to be within the top 10% most deleterious substitutions that can occur in the human genome; see **Online Methods** for the identification and definition of qualifying variants) within each gene were aggregated together for association analysis. Aggregation of variant counts was necessary for meaningful statistical comparisons between PACG cases and controls

✉e-mail: zhenxun.wang@duke-nus.edu.sg; khorcc@a-star.edu.sg

because most protein-altering variants are rare. We tested whether any of the ≈20,000 genes throughout the human genome bear an excess or deficit of qualifying variant burden in persons with PACG compared to unaffected individuals.

Significantly associated genes were evaluated for independent validation in an additional 2,519 patients with PACG and 471,724 unaffected individuals ascertained from 10 sites. Molecular biology experiments were performed to characterize one gene that surpassed genome-wide significance, and to assay the functional status of genetic variants. A de-novo validation incorporating the functional status of genetic variants was pursued in an independent confirmation study involving 208 PACG cases and 600 controls from Italy and Pakistan.

## Results

### Discovery exome sequencing analysis
From the discovery collection of 4,667 persons with PACG and 5,473 unaffected individuals (Supplementary Table 1), we identified 888,233 qualifying variants (see **Online Methods**) across 18,988 genes. Primary analysis revealed an exome-wide significant association at the gene *UBOX5* ($P = 1.03 \times 10^{-6}$). Persons with PACG were more likely to carry *UBOX5* qualifying variants compared to unaffected controls in each cohort of the meta-analysis (Odds Ratio, OR = 2.14, 95% confidence interval, ci = 1.57 – 2.89)(Fig. 1A, Supplementary Table 2). Because the association between rare variants at *UBOX5* and increased risk of PACG has not been previously reported, we evaluated if this association could have arisen due to confounding by population stratification. Projection of all carriers of *UBOX5* qualifying variants onto the top two principal components for genetic ancestry showed that the carriers were not biased along any major axes (Supplementary Fig. 1). Adjustment of the burden test by including the top 5 principal components of population stratification together with exome-wide variant count (see **Online Methods**) as continuous variables did not alter the exome-wide significant result (adjusted $P = 1.14 \times 10^{-6}$). These observations suggest that the association between carriers of rare *UBOX5* variants and increased risk of PACG was unlikely to be confounded by population stratification.

Secondary, sex-stratified analysis showed female carriers of *UBOX5* qualifying variants had nominally higher odds of PACG (OR = 2.23, 95%ci: 1.52-3.28) compared to male carriers (OR = 1.66, 95%ci: 0.95-2.91). Carriers of *UBOX5* qualifying variants had significantly elevated risk of acute primary angle closure subtype (OR = 2.39, 95%ci, 1.56 – 3.66), with female carriers having 3-fold increased odds (95%ci: 1.81-4.97) compared to female non-carriers (Fig. 1A).

### Replication exome sequencing analysis
To replicate the significant primary finding at *UBOX5*, ten additional case-control panels independent from the discovery study were evaluated. The first nine panels comprised 760 PACG cases and 3,844 unaffected individuals ascertained from hospital-based studies, and the tenth was from the community-based UK Biobank. In the UK Biobank, 469,639 participants underwent exome sequencing, of which 1,759 (0.37%) had PACG.

Persons with PACG were significantly more likely to carry *UBOX5* qualifying variants (17 carriers out of 760 PACG cases; 2.2%) compared to unaffected individuals (32 carriers out of 3,844 unaffected controls; 0.83%) in the nine hospital-based studies ($OR_{meta-analysis} = 5.5$, 95% ci: 2.5 – 11.9; $P_{meta-analysis} = 2 \times 10^{-5}$). This effect was also validated in the UK Biobank (24 carriers out of 1,759 persons with PACG [1.4%] compared to 3,864 carriers out of 467,880 unaffected individuals [0.83%], OR = 1.66, $P = 0.013$)(Fig. 1B).

A meta-analysis of all 7,186 participants with PACG and 477,197 unaffected individuals from the discovery and validation stages revealed a significant association between carriers of *UBOX5* qualifying variants and increased risk of PACG ($P = 1.25 \times 10^{-10}$; OR = 2.13, 95%ci, 1.69 – 2.69).

### Expression analysis of UBOX5
*UBOX5* encodes for U-box domain containing protein 5, a putative E3 ubiquitin-protein ligase[8]. In human eyes, positive staining for UBOX5 was observed in the iris sphincter pupillae (Supplementary Fig. 2), optic nerve head (Supplementary Fig. 3), axons of the retinal nerve fibre layer, retinal ganglion cells, inner and outer plexiform layers, amacrine cells in the inner nuclear layer, and photoreceptors (Supplementary Fig. 4). To confirm antibody specificity, we next evaluated UBOX5 expression in the eyes of mice using immunofluorescence. In the retina of mice, UBOX5 appeared to be expressed in the cytoplasm of Retinal Ganglion Cells (RGC). Within the Inner plexiform layer (IPL), the expression pattern highlights possible synaptic connections between the RGC, amacrine and bipolar cells in the IPL. These staining of UBOX5 observed in wild-type mice was not observed in the retina of UBOX5 knockout mice, thus validating the specificity of the UBOX5 antibody (Supplementary Fig. 5a). In the iris of mice, UBOX5 was positively expressed in the iris sphincter pupillae and some blood vessel muscles of the iris, as well as in the lens epithelium. Expression of UBOX5 was abolished in the anterior chamber tissues of UBOX5 knockout mice thus validating the specificity of the UBOX5 antibody (Supplementary Fig. 5b). The expression patterns of UBOX5 in human and mice eyes were remarkably similar and suggest that the iris sphincter pupillae and retina ganglion cells could be sites of pathology in PACG.

### Discovery of BIP as a substrate of UBOX5
To identify target proteins of UBOX5, we engineered a FLAG-tagged UBOX5-ubiquitin binding domain (UBD) fusion protein construct that traps proteins ubiquitinated by UBOX5[9,10] (Supplementary Methods and Supplementary Fig. 6). We introduced this construct in human embryonic kidney 293 cells and then retrieved the proteins bound to it. Using mass spectrometry, we analyzed which proteins interacted with UBOX5. Across two independent experiments, three heat shock proteins (HSP) (HSPA8, HSPA1B, and HSPA5) were among the most enriched and highest scoring proteins binding to UBOX5-UBD (Supplementary Table 3). We prioritized HSPA5 (which encodes for binding immunoglobulin protein, BIP) for further characterization due to its role as a sensor[11] and inducer of the unfolded protein response (UPR) pathway[12] via endoplasmic reticulum (ER) stress[13]. Abnormalities in the UPR pathway are linked to glaucomatous optic neuropathy[14]. The other two HSPs were not followed up as they have different cellular functions[15].

To validate our initial finding, the experiment was repeated by immunoprecipitation of the fusion protein construct in cells expressing both FLAG-tagged UBOX5-UBD and HA-tagged ubiquitin. The first round of immunoprecipitation with anti-FLAG antibody showed BIP binding to the UBOX5-UBD fusion protein (Fig. 2a). Eluates were then subjected to a second round of immunoprecipitation with anti-HA antibodies which yielded ubiquitinated BIP (Fig. 2a), suggesting that UBOX5 has bona-fide ubiquitin ligase activity. In support of our observations that UBOX5 could ubiquitinate BIP, we also observed binding of native UBOX5 to BIP (Fig. 2a).

To directly test UBOX5's ability to ubiquitinate BIP, we introduced MYC-tagged BIP, HA-tagged ubiquitin, and native UBOX5 into cells, followed by immunoprecipitation of BIP with anti-MYC antibody and immunoblotting with anti-HA antibodies. We observed that BIP was substantially ubiquitinated only when UBOX5 was present (Fig. 2b). Ubiquitinated BIP abundance did not increase with pre-treatment with proteasome inhibitor; analysis of mobility shift between unmodified and ubiquitinated BIP in immunoblots also revealed that mono-ubiquitinated BIP appeared to be the dominant species when UBOX5 was co-expressed (Fig. 2b), suggesting that the ubiquitination event did not lead to degradation of BIP. We also found that endogenous UBOX5 expression was induced by ER stress (Fig. 3a). Given that most protein translation is strongly inhibited under conditions of ER stress[16], the observed induction of UBOX5 expression strongly suggests that UBOX5 could be involved in modulating BIP under these conditions.

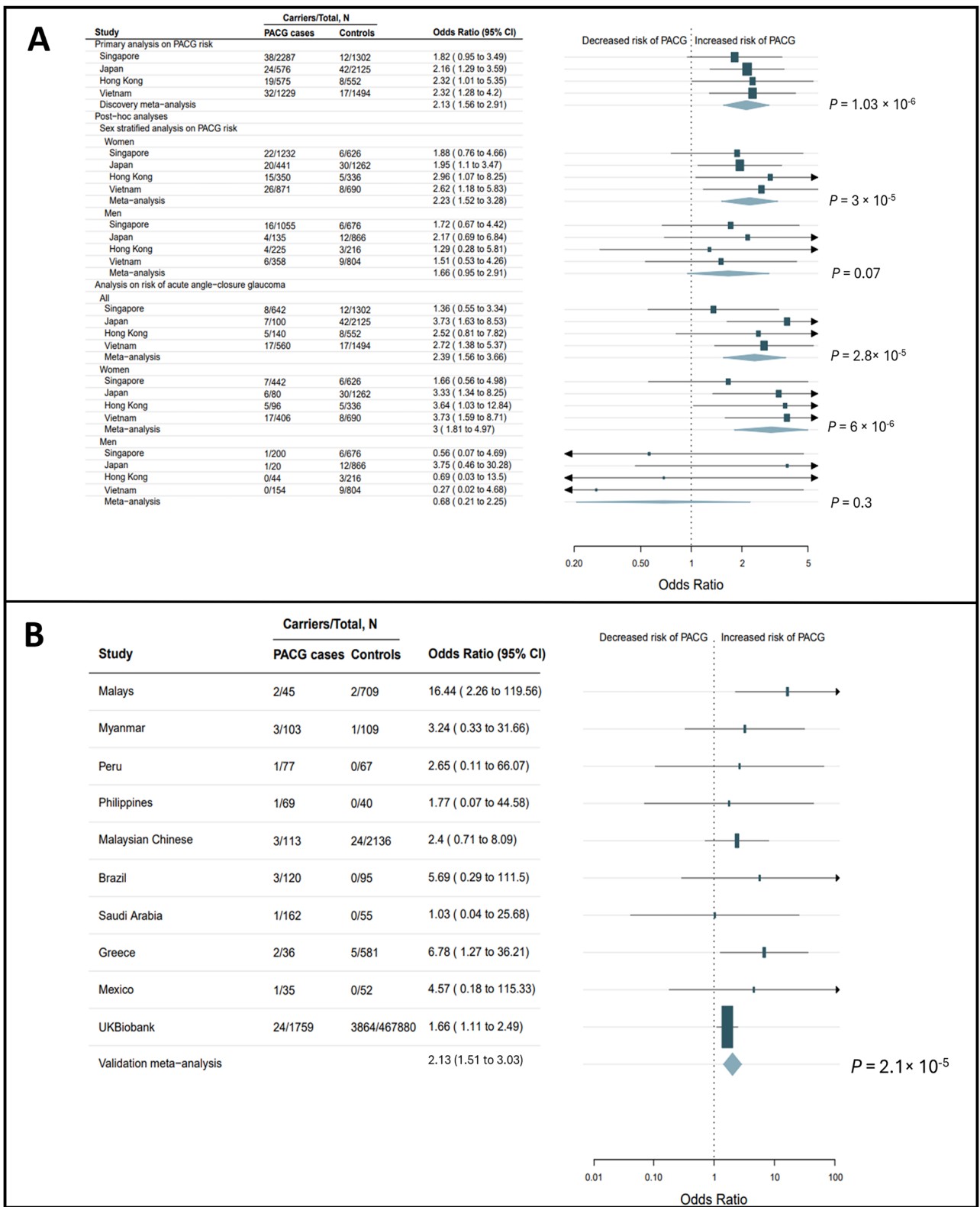

**Fig. 1 | *UBOX5* is associated with increased risk of primary angle-closure glaucoma (PACG).** Cochran Mantel-Haenszel stratified meta-analysis was used to calculate an overall odds ratio for the study cohorts in the discovery (**a**) and validation (**b**) stage. Data are presented as Odds Ratios with accompanying 95 percent confidence intervals. *P*-values are two-sided, with no adjustments for multiple comparisons. **A** Forest plots for the discovery exome sequencing stage (comprising 4,667 persons with PACG and 5,473 unaffected individuals enrolled from 4 sites). Primary analysis describing the association between carriage of rare, protein-altering variants at *UBOX5* and risk of PACG are shown together with post-hoc analyses stratifying by sex, as well as by presence of acute primary angle closure. **B** Forest plot from the primary analysis of the validation stage (comprising 2,519 affected individuals and 471,724 unaffected individuals enrolled from 10 sites).

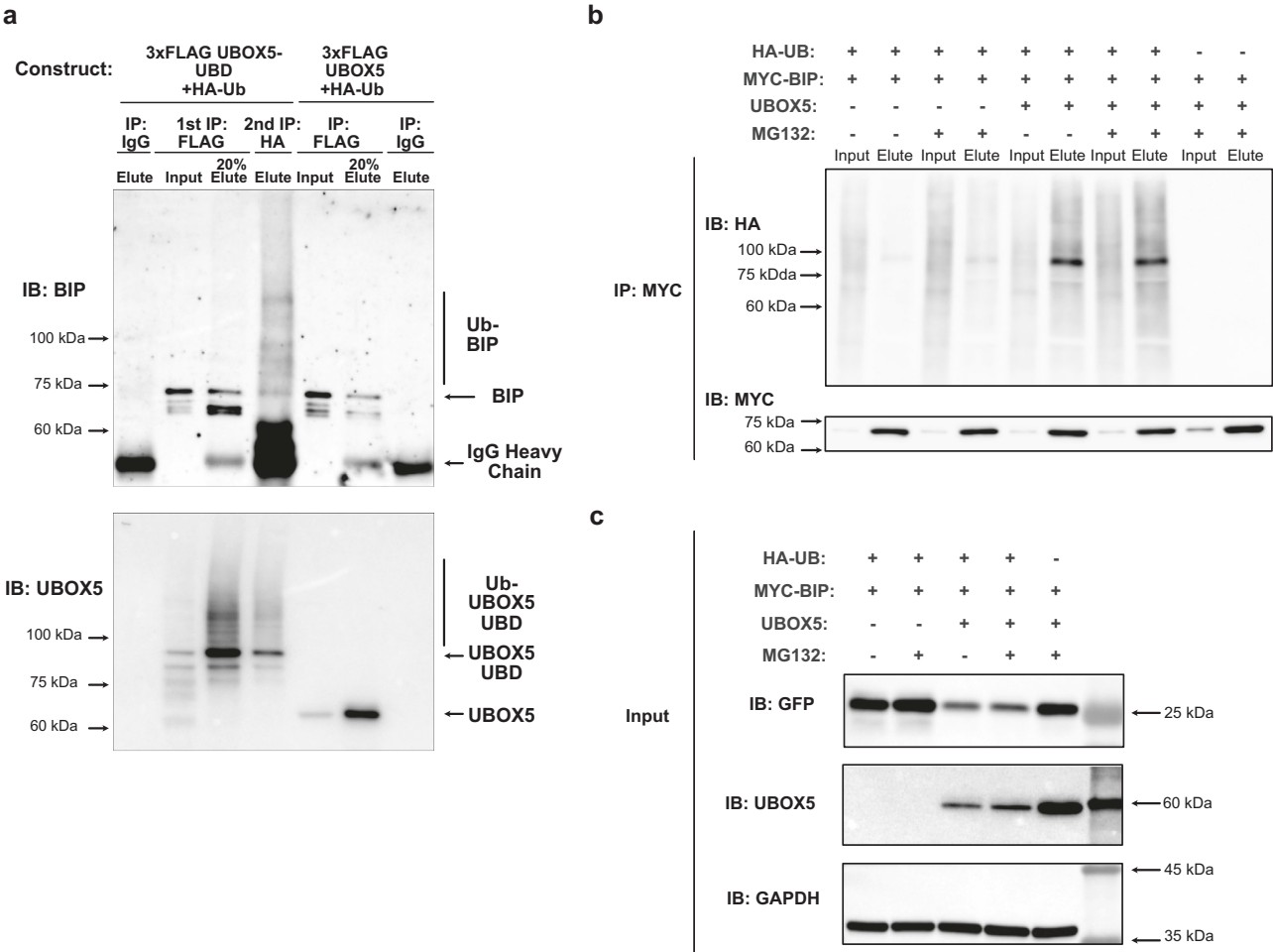

**Fig. 2 | UBOX5 has E3 ubiquitin ligase activity. a** A UBOX5 chimeric construct (UBOX5-UBD) where the UBOX5 open reading frame was fused by a flexible linker to a FLAG-tagged ubiquitin binding domain was generated. FLAG-tagged UBOX5 (3 lanes on the right) or FLAG-tagged UBOX5-UBD construct (4 lanes on the left) and HA-tagged ubiquitin was co-transfected into HEK293 cells. After the first immunoprecipitation of lysates by anti-FLAG antibody, 20% of eluates was kept for analysis. Eluate from the UBOX5-UBD was further immunoprecipitated with anti-HA antibody to enrich for ubiquitinated proteins. To serve as antibody specificity control (negative controls), lysates were mock immunoprecipitated with mouse immunoglobulin. Immunoblotting of BIP was performed on inputs and eluates as indicated. Bands corresponding to ubiquitinated BIP and ubiquitinated UBOX5-UBD chimeric protein are indicated by vertical lines, while the unmodified proteins are indicated by arrows on the right. Positions of the molecular weight markers are indicated by arrows on the left. IB: FLAG and IB:HA are shown in Supplementary Fig. 13. The experiment was repeated independently 3 times. Source data are provided as a Source data file. **b** MYC-tagged BIP, empty vector, UBOX5, or HA-tagged ubiquitin was co-transfected into HEK293 cells in the indicated combinations. 24 h later, cells were treated with 0.7 uM Tharpsigargin for 16 h, and then further treated with MG132 (a proteasomal inhibitor) for 6 h as indicated. Cells were then harvested and a MYC immunoprecipitation was performed on the input lysates. Eluates were immunoblotted with antibodies against HA to assess the extent of BIP ubiquitination. The membrane was then stripped and a MYC immunoblot was performed to assess immunoprecipitation efficiency. Ubiquitination of BIP was only observed when UBOX5 was expressed. The degree of ubiquitination of BIP did not appear to differ with the addition of MG132, a proteasomal inhibitor, suggesting that the ubiquitinated BIP was not degraded by the proteasome pathway. The experiment was repeated independently 3 times. Source data are provided as a Source data file. (**c**) UBOX5 or its empty vector contains a GFP open reading frame, which allows for assessment of transfection efficiency by assessing GFP abundance in input lysates. Human UBOX5 immunoblots were used to verify expression of UBOX5. GAPDH was used as loading control. The experiment was repeated independently 3 times. Source data are provided as a Source data file.

Cell fractionation of UBOX5-expressing HEK293 cells demonstrate that UBOX5 is present in the ER together with BIP (Fig. 3b), suggesting that UBOX5 could be directly involved in ER stress response via interacting with BIP (Fig. 2a). Although we also found that ubiquitination of BIP by UBOX5 did not depend on ER stress (Fig. 3c), subsequent experiments were carried out in ER stress conditions to ensure physiological relevance, as ER stress is the cellular condition where both BIP and UBOX5 were induced.

### Functional characterization of *UBOX5* variants

We assessed the ability of *UBOX5* genetic variants to ubiquitinate BIP (**Supplementary Methods** and Supplementary Fig. 7). In all, 35 UBOX5 variants emerging from the discovery and UK Biobank cohorts were tested (Supplementary Table 4). These 35 variants accounted for 89

percent of *UBOX5* carriers in PACG cases and 62 percent of *UBOX5* carriers in unaffected individuals. Laboratory tests suggest 20 of the 35 variants to be functionally impaired (defined as having functional activity significantly below that of wild-type UBOX5 activity; Fig. 4 and Supplementary Fig. 8). Nineteen of the 20 (95%) functionally impaired variants appeared to be carried more often by persons with PACG compared to by unaffected individuals. Conversely, 10 of the 15 (66.7%) variants with normal function were carried more often by unaffected individuals compared to affected individuals (Supplementary Table 4).

The performance of bioinformatic algorithms used to functionally classify variants was compared to laboratory tests. Out of the 35 *UBOX5* variants tested, 33 were missense substitutions and returned predictions from both CADD[17] and Polyphen2[18] algorithms (Supplementary

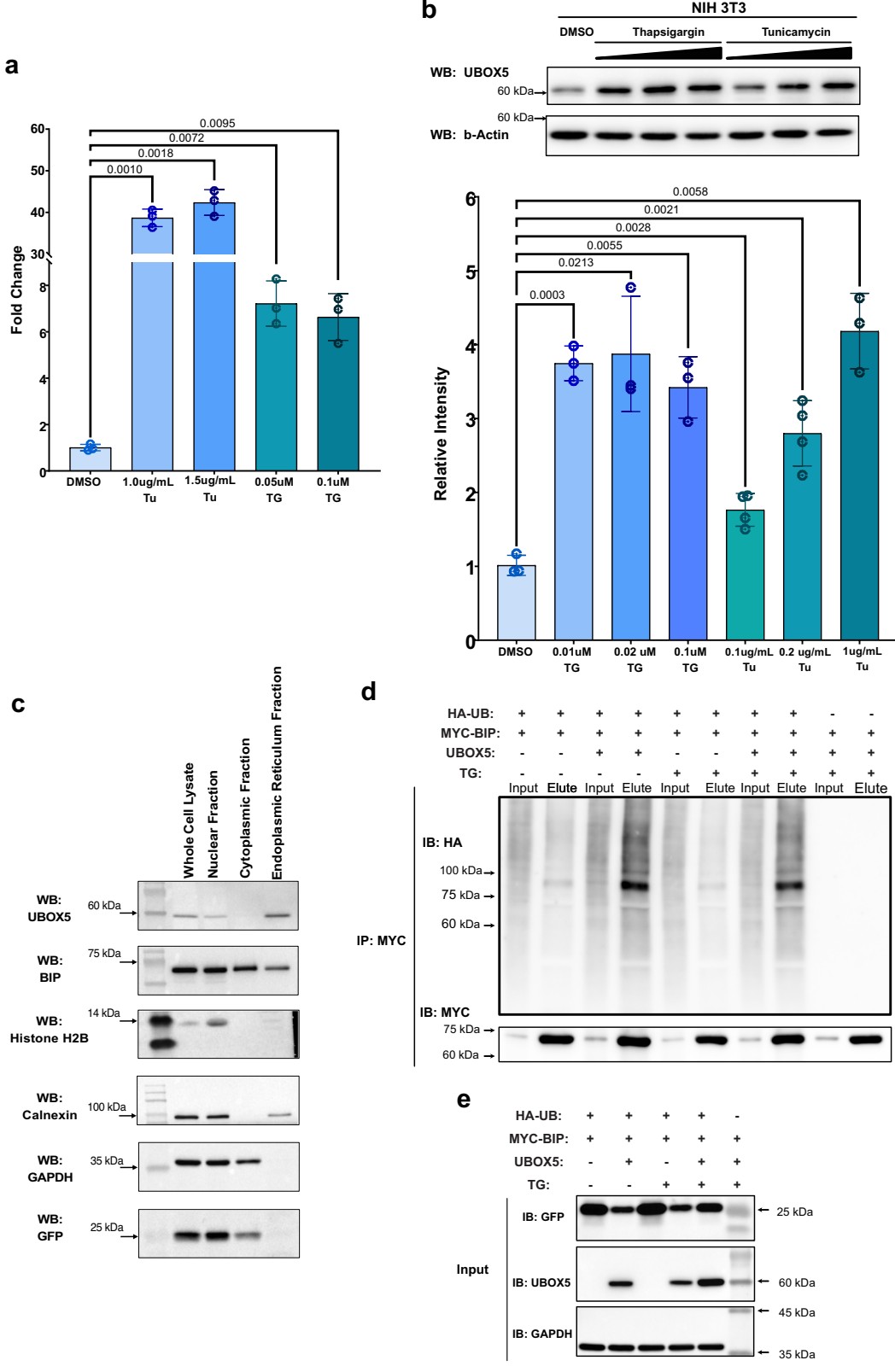

**Nature Communications** | (2025)16:7620

Table 4). The CADD algorithm correctly classified 18 out of the 20 (90%) functionally impaired variants, but only correctly classified 3 out of the 15 (20%) functionally normal variants (Supplementary Table 5), a difference which was statistically significant ($P = 1.5 \times 10^{-5}$). Polyphen2 appeared to have lower success in classifying both functionally impaired variants (65%, or 13 out of 20) and functionally normal variants (46.7%, or 7 out of 15; $P = 0.3$ for difference).

We next asked whether wild-type UBOX5 might exert its effect by altering the half-life of BIP since it was recently demonstrated to be a short-lived protein[19]. To this end, pulse-chase analysis of BIP turnover in the presence or absence of UBOX5 expression was performed in HEK293 cells after induction of the unfolded protein response with tharpsigargin. The presence of wild-type UBOX5 significantly increased the half-life of BIP compared to absence of UBOX5 in the

**Fig. 3 | The biological properties of UBOX5. a, b** UBOX5 and BIP are both induced by ER stress. **a** NIH3T3 cells were treated with the indicated ER stress inducers tunicamycin (Tu) and thapsigargin (Tg). (Upper) Endogenous UBOX5 mRNA abundance was quantified by qPCR in biological triplicates, normalised against mouse beta-actin transcript. Relative fold change of transcript is reported against control DMSO treatment. Error bars represent standard deviation. (**b**, Top) Protein abundance of endogenous mouse UBOX5 in ER-stressed NIH3T3 cells. Beta-actin was used as loading control. Of note, UBOX5 mRNA and protein is induced in response to ER stress. (**b**, Bottom): Densitometric quantitation of UBOX5 bands. UBOX5 band intensity of each sample is normalised to corresponding beta-actin intensity. The amount of TG or TU used is indicated on the *x*-axis. *P*-values were generated from two-sided Welch's *t*-test. Error bars represent standard deviation. Source data are provided as a Source data file. **c** Cellular Localization of UBOX5: Immunoblots of HEK293 cells transiently transfected with UBOX5 expression plasmid and treated with 0.7 uM Thapsigargin for 16 h. Cellular fractions are indicated above. Whole cell lysates were fractionated into cytoplasmic, endoplastic

reticulum (ER) fractions and nuclear fractions by stepwise centrifugation. Indicated antibodies are shown. Positions of molecular weight markers are indicated on the left with arrows. GFP is used as transfection control, Calnexin is used as fractionation control for ER and nuclear fraction; Histone H2B is used as fractionation control for nuclear fraction. GAPDH is used as fractionation control for cytoplasmic and nuclear fractions. The experiment was repeated independently 3 times. Source data are provided as a Source data file. **d, e** The ability of UBOX5 to ubiquitinate BIP is not dependent on cellular stress. **d** MYC-tagged BIP, empty vector, UBOX5, or HA-tagged ubiquitin was co-transfected into HEK293 cells in the indicated combinations. 24 hours later, cells were treated with 0.7 μM Thapsigargin (TG) or DMSO for 16 h. TG is a known inducer of endoplasmic reticulum stress. Cells were then harvested and a MYC immunoprecipitation was performed on the input lysates. Eluates were immunoblotted with antibodies against HA to assess the extent of BIP ubiquitination. **e** UBOX5 or its empty vector contains a GFP open reading frame, which allows for assessment of transfection efficiency by assessing GFP abundance in input lysates. GAPDH was used as loading control.

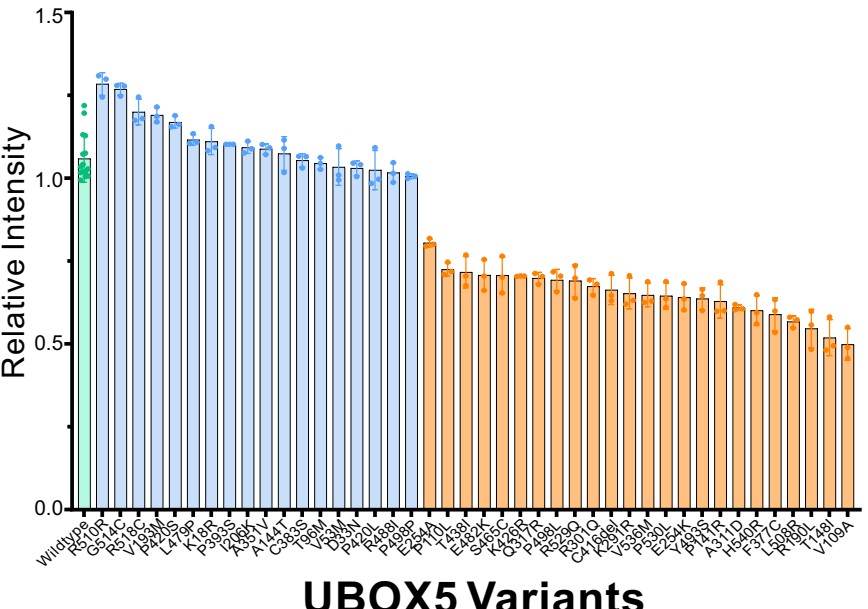

## UBOX5 Variants

**Fig. 4 | Gel quantitation data measuring the functional activity of UBOX5 protein-altering variants to ubiquitinate BIP.** Data are presented in biological triplicates. Intensity of the HA band was quantified and then normalized to the MYC signal of the same elute sample in the same gel. MYC signal was obtained after stripping the initial HA immunoblot and re-probing with MYC antibody. Wild-type UBOX5 (green bar) was measured independently 15 times to provide a confidence estimate of wild-type UBOX5 activity, so that functionally deficient UBOX5 variants

(defined as <80% activity of wild-type UBOX5) can be more confidently identified. Variants in blue bars represent normally functioning UBOX5 alleles. A total of 42 UBOX5 variants were tested, comprising 35 variants detected in the initial study, and 7 variants in the validation study from Italy and Pakistan. Representative Western Blot gel photographs are presented as Supplementary Fig. 7. Error bars represent standard deviation. Source data are provided as a Source data file.

presence of thapsigargin-induced ER stress, but not in baseline conditions without ER stress (Fig. 5a). We next selected 4 *UBOX5* variants, comprising 3 that were functionally deficient (K291R, R301Q, and S465C) and one that had normal function (D33N) for further analysis on their potential impact on BIP half-life. While the presence of D33N was observed to increase the half-life of BIP to a similar extent as wild-type UBOX5, all 3 functionally deficient UBOX5 variants failed to do so (Fig. 5b). We note that mutations present in 2 out of the 3 functionally deficient variants (K291R and R301Q) were in the UBOX domain of UBOX5, an essential domain for ubiquitination activity of mammalian UBOX proteins[20].

### De-novo validation in additional PACG cases and controls

We sought further confirmation of the previous findings in a de-novo validation study. Two panels from Italy and Pakistan were available, where complete eye examinations were performed for ascertaining of

participant case-control status. This approach allowed for a fresh re-evaluation of the association between functional status of *UBOX5* variants and risk of PACG. Using whole-exome sequencing, the coding-sequence of *UBOX5* was surveyed in 70 persons with PACG and 242 unaffected controls from Italy, and 138 persons with PACG and 358 unaffected controls from Pakistan. Fourteen unique variants were detected, of which 7 were newly observed (p. Arg518Cys, p. Arg510Arg, p. Leu508Arg, p. Pro393Sser, p. Val193Met, p. Thr148Ile, and p. Arg529Gln; Supplementary Table 6). These were subjected to functional laboratory testing in the same manner as the previous 35 variants (Fig. 4).

Of the 7 newly observed variants, 3 (p. Arg529Gln, p. Leu508Arg, and p. Thr148Ile) showed clear evidence of functional deficiency (Fig. 4), and all 3 were present exclusively in persons with PACG (Supplementary Table 6). Statistical analysis suggests carriers of these functionally deficient *UBOX5* variants had 12.1-fold increased odds of

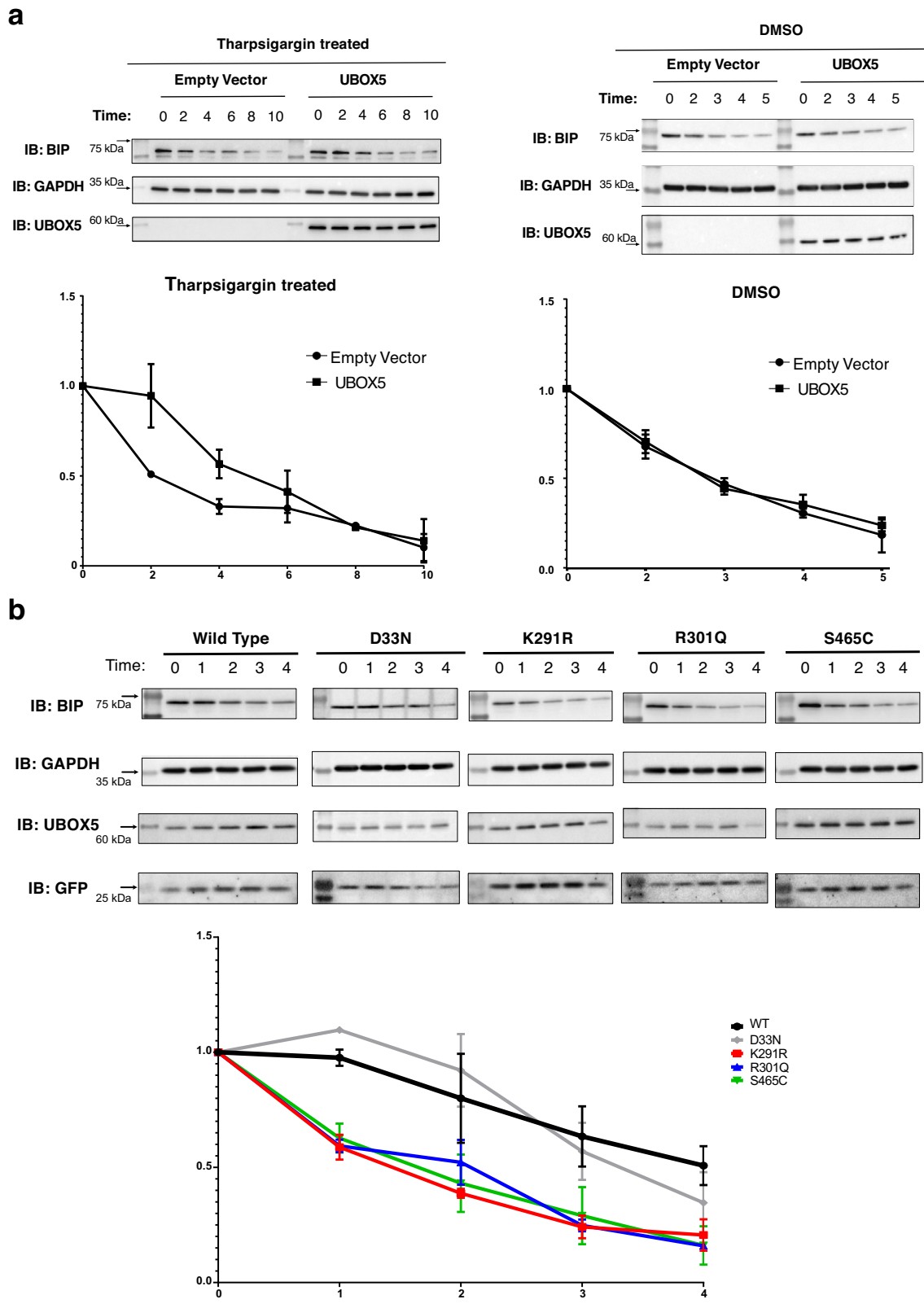

PACG (95%ci, 2.2 – 66.4; *P* = 0.002) compared to non-carriers. In contrast, no statistically significant association between odds of PACG and *UBOX5* qualifying variants as defined by CADD scores (OR = 1.36, 95%ci, 0.35– 5.33) or by Polyphen2 predictions (OR = 2.28, 95%ci, 0.54–9.72) were detected.

## Discussion

This whole-exome sequencing study of >480,000 participants suggest *UBOX5* to be a risk gene for PACG. Biological characterization of *UBOX5* variants showed functionally deficient alleles to be enriched in persons with PACG compared to controls. UBOX5 functions to mono-

**Fig. 5 | UBOX5 affects BIP half-life. a** Wild-type UBOX5 increases the half-life of BIP in the presence of thapsigargin-induced ER stress. Top left panel: Pulse chase of BIP with or without expression of UBOX5 in HEK293 cells. HEK 293 cells were transfected with UBOX5 or empty vector. 24 h later, cells were treated with a pulse of 0.25 μM Tharpsigargin for 2 hours, and cycloheximde (150 mM) was added. Cells were collected at indicated time points for immunoblotting. GAPDH was used as loading control. Positions of molecular weight standards are indicated on the left. Top right panel: The same experiment repeated with Tharpsigargin substituted with DMSO carrier control. Bottom panels: Quantitation of BIP band intensities normalized to GAPDH intensity for every indicated time point. Separate graphs were shown for cells treated with Tharpsigargin (Left) or DMSO control (Right). Intensities are shown as fold changes compared to normalized BIP intensity at $t = 0$ of pulse chase. Data are presented as mean values +/- standard deviation. Error bars indicate standard deviation; 3 biological replicates were used for quantitation.

Source data are provided as a Source data file. **b** Variant UBOX5 and their effects on the half-life of BIP. Top: Pulse chase of BIP in the presence of wildtype UBOX5 or variant UBOX5 (D33N, K291R, R301Q, and S465C) in HEK293 cells. HEK 293 cells were transfected with wildtype UBOX or indicated variants. 24 h later, cells were treated with a pulse of 0.25 μM Tharpsigargin for 2 h, and cycloheximde (150 mM) was added at time = 0. Cells were then subsequently collected at indicated time points (in hours) for immunoblotting. GAPDH was used as loading control. UBOX5 expression was verified as indicated. GFP, expressed from a separate locus in the vector used, was also used to verify success of transfection. Bottom: Densitometric quantitation of BIP band intensities in cells expressing indicated UBOX5 variants, normalized to GAPDH intensity for every time point on the same blot. Intensities are shown as fold changes compared to normalized BIP intensity at $t = 0$ of pulse chase. Three biological replicates were analyzed. Error bars represent standard deviation. Source data are provided as a Source data file.

ubiquitinate BIP. This process may modulate BIP's interactions[21], stabilize and protect it from poly-ubiquitination and proteasomal degradation[22], or alter its cellular localization[23].

Although little is known about both UBOX5 and BIP in the eye, our data reveals the following molecular clues. Firstly, the ubiquitination of BIP by UBOX5 appeared to increase the half-life of BIP. BIP senses and responds to ER stress by inducing the unfolded protein response (UPR) pathway. This pathway is conserved in mammalian cells[24] and in plants[25], and cells with pathway deficiencies were significantly less able to survive stressful stimuli, suggesting that activation of the UPR pathway in response to ER stress confers cytoprotective properties. Protein abundance of BIP is induced during ER stress to increase cellular protein folding capacity. Increasing its half-life would thus further increase cellular protein folding capability under these conditions, and lead to a rapid resolution of ER stress and return to homeostasis. Second, UBOX5 is expressed in the iris, and particularly in the pupil sphincter, the primary site of physiological dysfunction in PACG. Increasing age is a known risk factor for PACG[26], and the mechanical properties of the iris and pupil sphincter might be affected by aging related alterations. Reduced cellular protein folding capacity is a characteristic of aging cells[27], and decreased UBOX5 function due to carriage of functionally deficient UBOX5 genetic variants could aggravate this condition. We speculate that such a situation may associate with an increased probability of precipitating 'pupil block', one of the mechanisms underlying PACG[28]. Carriers of *UBOX5* variants had elevated odds of acute primary angle-closure (Fig. 1A), an ocular emergency caused by acute pupil block.

The high intraocular pressure in patients with PACG triggers endoplasmic reticulum (ER) stress in retinal ganglion cells[14,29–33]. Experimental modulation of ER stress appeared to protect retinal ganglion cells from cell death in a mouse model of glaucoma[34]. As we also observed UBOX5 was also expressed in retinal ganglion cells, its deficiency might lead to increased susceptibility to glaucomatous optic neuropathy. Our experimental observation showing UBOX5 increasing the half-life of BIP (which has cytoprotective properties) supports this hypothesis, as this could increase cellular capability for protein folding during ER stress conditions[35].

This study has several limitations. First, the case control study design precluded assignment of causal relationships. Second, in keeping with published observations, we found PACG to be more prevalent in female participants compared to male participants. The odds ratio of UBOX5 carriers were observed to be higher in females compared to males. Out of the 4,667 persons with PACG that were analyzed in the discovery exome sequencing study, only 1,773 were men (38 percent), and out of the 1442 persons with acute angle-closure glaucoma, only 418 were men (29 percent). Thus, the substantially lower number of men with PACG may suggest lower statistical power to detect an effect, rather than it being a true sex-specific effect. Third, because the discovery analysis was performed in participants of Asian

ancestry but validation pursued in diverse panels of participants from around, pinning down the series of UBOX5 alleles that are causally related to PACG risk will require additional follow up. The current study size does not provide sufficient statistical power to dissect the contribution of ancestry-specific UBOX5 alleles to PACG risk. Fourth, because the determination of PACG status was based on diagnosis codes in the UK Biobank, phenotype misclassification may be greater than that which would be seen in a specialist-adjudicated collection enrolled in hospital-based settings. Nonetheless, the successful validation of the association between carriers of *UBOX5* variants and increased risk of PACG even in the face of such limitations in the UK Biobank cohort may serve to strengthen the conclusions of the study.

In summary, carriage of functionally impaired *UBOX5* variants was associated with a substantial increased risk of PACG. Further research is needed to understand how deficiencies in the UBOX5−BIP signalling pathway might result in the development of PACG.

## Methods
### Study design
The study design and conduct complied with all relevant regulations regarding the use of human study participants and was conducted in accordance with the criteria set by the Declaration of Helsinki. All participants were recruited after written informed consent. Detailed descriptions of each participating site are included in Supplementary Methods. The studies were approved by all relevant local and hospital Institutional Review Boards of the participating sites, and are described in Supplementary Methods. This genetic association study comprised independent discovery, validation, and de-novo confirmation stages (Supplementary Table 1).

The inclusion criteria for patients with PACG were:
a) Patients with previous APAC and/or
b) Patients with chronic PACG (defined below)

Patients with APAC were defined by the presence of at least two of the following symptoms: ocular pain, nausea and / or vomiting, with an antecedent history of blurring of vision; a presenting IOP of >28 mmHg on Goldmann applanation tonometry; and the presence of at least three of the following signs: conjunctival injection, corneal edema, mid-dilated non-reactive pupil, and a shallow anterior chamber.

Patients diagnosed with chronic PACG had asymptomatic closure of the angle assessed by gonioscopy, accompanied by glaucomatous optic neuropathy (GON), defined as abnormally large optic disc excavation and loss of neuro-retinal rim tissue with a vertical cup-to-disc ratio greater than the 97.5 percentile of the population when examined with a 78D bio-microscopic lens. A GON diagnosis was confirmed by the presence of visual field loss (consistent with glaucoma as per International Society of Geographical and Epidemiological Ophthalmology criteria) detected with static automated white-on-white

threshold perimetry (program 24-2 SITA, model 750, Humphrey Instruments, Dublin, CA).

The unifying characteristic of patients with APAC (acute primary angle-closure) and PACG (primary angle closure glaucoma) is that the cause of their angle closure was not secondary to any other pathology (hence classified as primary angle closure), and that all patients have gonioscopic angle closure with significant glaucoma disease severity. Patients younger than 50 years were excluded, as were patients with secondary forms of angle closure glaucoma such as neovascular glaucoma. As far as possible, unaffected control individuals were participants ≥50 years old and an eye examination to confirm the absence of glaucoma. Participants to this study were not compensated.

## Whole exome sequencing
Genomic DNA from all participants was extracted from venous blood. Whole-exome sequencing libraries were prepared and sequenced as previously described[36].

## Bioinformatic analyses after sequencing
Bioinformatics procedures for the mapping of sequencing reads, identification of sequence variants, annotation of variants, and quality control on individual variants and individual samples were applied uniformly on all samples blinded to case-control status.

Raw 2×151 base-pair paired end DNA reads from Novaseq 6000 instruments were aligned to the hg19 genome build. After alignment, reads that were found to duplicate the start position of another read were flagged as 'duplicates' and excluded from further analysis. We performed variant detection from the aligned reads using exons and 50 base-pair regions flanking the exons. To avoid erroneous findings, we processed all data uniformly from the beginning by aggregating BAM files from unaligned sequence reads using PICARD, Burrow-Wheeler Aligner (BWA), and Genome analysis tool kit (GATK) software packages, following best practice guidelines (https://www.biorxiv.org/content/10.1101/201178v3).

Genetic variants were identified and genotyped using recalibrated BAM files using Haplotype caller from GATK. Sequence variants were first identified and genotyped in batches of 200 samples per batch. For the discovery exome sequencing collection, joint calling was then conducted for the 4,667 persons with PACG and 5,473 unaffected individuals from Singapore, Hong Kong, Vietnam, and Japan. The same method for variant identification and joint calling was used in the first validation stage comprising 760 patients with PACG and 3,844 unaffected control individuals who were whole-exome sequenced from 9 hospital-based collections, as well as in the 208 participants with PACG and 600 unaffected controls from Italy and Pakistan.

Exome sequencing from the UK Biobank (1,759 participants with PACG and 467,880 unaffected controls) was performed using the xGen probe library from Integrated DNA Technologies (IDT). The multiplexed samples were pooled and then sequenced using 75-base-pair paired-end reads with two 10-base-pair index reads on the Illumina NovaSeq 6000.

## Variant level quality control
Strict quality control measures were implemented to ensure that only analyzed high quality sequence variants were analyzed. All identified genetic variants must be supported by a sequencing depth of at least 10X coverage. To be included for further analysis, the call rate for each variant must be > 95%. All samples were then stratified according to country of recruitment and variants showing significant deviation from Hardy-Weinberg Equilibrium ($P_{for\ deviation} < 0.0001$) were excluded. All variants showing statistically significant differences in genotyping completion rate between cases and controls ($P_{for\ difference} < 0.001$) were also excluded. We treated multi-allelic variants as collections of independent bi-allelic variants. For multi-allelic variants, the variant was excluded if any of the individual alleles failed any quality control measure. To minimize falsely identified genetic variants, all variants were required to have an allelic balance (AB) of 20% < AB < 80%.

All identified variants passing quality checks had variant quality score log-odds (VQSLOD) score corresponding to truth set sensitivity of 99.4%. These scores were derived from the variant quality score recalibration (VQSR) step from the GATK workflow package.

## Sample level quality control
Stringent quality control checks were applied to remove outlying samples that may bias association tests and result in false-positive associations. Each sample qualifying for downstream analysis must have > 80% of targeted bases sequenced to 20X or more. For this study, we observed >90% of targeted bases sequenced to 10X or more. The checks for each participating country (Singapore, Hong Kong, Vietnam, and Japan) were performed separately, and samples showing an extreme distribution of heterozygous genotypes (which could reflect samples that had cross-contamination) were excluded from analysis. Outlying samples by ancestry principal component analysis were similarly excluded, as were sample duplicates and related samples. We required samples to have a completion rate of ≥95%.

We excluded from further analysis samples with an unusually low number of singletons as these also suggest sample cross-contamination. We also excluded samples with an unusually high number of singletons as this metric suggests low quality DNA (Supplementary Fig. 9).

## Identification of qualifying genetic variants for association analysis with PACG status
Technical details on variant annotation are appended in **Supplementary Methods**. Rare variants predicted to impair protein function are henceforth referred to as 'qualifying variants', and were identified and defined using the following criterion:

a) Minor allele frequency <1% across all ancestry groups studied.
b) Variants that disrupt the protein-coding sequence (stop-gained, start loss, frameshift, or canonical splice-site mutations).
c) Missense variants with a Combined Annotation Dependent Depletion (CADD) scaled score of >10. CADD > 10 reflects variants predicted to be within the top 10% most deleterious substitutions that can occur in the human genome[17].

## Ancestry principal component calculation for whole exome sequenced samples
For each sample, ancestry principal component scores were calculated using a genome-wide profile of ancestrally informative common genetic variants that were not in linkage disequilibrium with one another (defined as pair-wise $r^2 < 0.1$). All ancestrally informative genetic variants used to calculate principal component scores also fulfilled strict quality control criteria for genotyping completion rate and non-significant deviation from Hardy-Weinberg equilibrium. Projection of all samples onto a plot showing the top two ancestry principal components showed excellent genetic matching between PACG patients and unaffected control (Supplementary Fig. 10).

## Sanger Sequencing check to verify rare allele calls
Twenty-eight participants carrying *UBOX5* qualifying variants identified by high-throughput exome sequencing were selected for verification of carrier status using Sanger capillary sequencing. We observed all *UBOX5* variant calls to be validated across all samples. We append specimens of the Sanger sequencing traces in Supplementary Fig. 11.

## Statistical analysis
In the discovery study design, primary analysis evaluating the association between genetic variants and presence of primary angle-closure glaucoma (PACG) was performed on all participants without consideration of sex. We used the Cochran Mantel-Haenszel stratified

meta-analysis to summarize exome-wide gene-based burden tests for the samples from Singapore, Hong Kong, Japan, and Vietnam. In previous studies, this method is robust to low or even zero counts often seen in rare variant burden tests[37]. Exome-wide significance was specified at $P < 2.5 \times 10^{-6}$ to reflect multiple-testing correction for ≈20,000 genes. The same test was used to summarize gene-based burden tests for the validation and confirmation studies. All $P$-values reported were two-tailed. For the calculation of odds ratios with zero cells counts, the Haldane-Anscombe correction was applied[38].

## Statistical power calculation

Our study design had an estimated statistical power of 88% to surpass exome-wide significance (preset as $P < 2.5 \times 10^{-6}$ to account for gene-based burden tests on the ≈20,000 genes found in the human genome) in the discovery exome sequencing analysis. This power calculation applies to all genes with cumulative rare variant burden of at least 1 percent, with a differential variant burden between cases and controls associated with an odds ratio (OR) of at least 2.0 (Supplementary Table 7). Considering other scenarios, statistical power remains adequate for discovering genes at the exome-wide significance threshold for which either the differential variant burden was associated with a large effect size (e.g. ORs ≥ 2.5 could be detected for genes with cumulative rare variant burden as low as 0.5%, at 86.8% power), or genes for which large numbers of qualifying rare variant carriers are observed (e.g. genes with a minimum cumulative rare variant burden of 2% could be detected with an OR ≥ 1.7 at >90% power).

## Additional controls for experimental and analytical biases

We tried to ensure that qualifying variants are identified equally in both the case (persons with PACG) and control (unaffected individuals) groups. This was achieved by deliberately avoiding systematic differences in the manner which DNA samples from the case and control groups are processed and analyzed.

First, we required that sequencing coverage of the target regions between persons with PACG and unaffected controls be equal. This could be evaluated by plotting exome-wide variant count profiles between PACG cases and unaffected controls and ensuring that they are not markedly different from one another (Supplementary Fig. 12). Second, we ensured that DNA from persons with PACG and DNA from controls were prepared using the same exome capture kit as far as possible. Third, sequencing reads must be mapped in a manner blinded to case-control status of the participants. Fourth, variant calling was also performed blinded to case-control status.

### UBOX5 biological assays

Detailed experimental descriptions for UBOX5 biological assays can be found in Supplementary Methods.

## Induction of UBOX5 expression

NIH 3T3 cells (CRL-1658, purchased from the American Type Culture Collection, ATCC) were plated at $8 \times 10^5$ cells per well of a 6 well plate and then treated with Thapsigargin (#sc-24017, Santa Cruz) or Tunicamycin (#sc-3506A, Santa Cruz) for 16 h, before being lysed with RIPA buffer (#89900, Thermo Scientific). Lysates were immunoblotted with antibodies against mouse UBOX5 (Genemed) and GAPDH (#sc-47724, Santa Cruz) as a housekeeping control.

## Substrate trapping experiments

To identify all possible ubiquitination substrates of UBOX5, a UBOX5 – ubiquitin binding domain (UBD) fusion plasmid construct was generated, which was then transfected into human embryonic kidney (HEK)-293 cells (CRL-3216) purchased from ATCC. Harvested cells were processed for immunoprecipitation assays before mass spectrometry analysis.

## Mass spectrometry analysis

Immunoprecipitated samples were digested with trypsin and analysed by liquid chromatography with tandem mass spectrometry (LC-MS/MS). MS data was analysed using Proteome Discoverer 2.4, and proteins were identified at 1% false discovery rate ($q$-value ≤ 0.01) with a minimum of two unique peptides per protein.

## Tandem immunoprecipitation for UBOX5 and BIP

HEK293 cells were transfected with FLAG-tagged UBOX5-UBD construct along with hemagglutinin (HA)-tagged ubiquitin. The FLAG tag is a specific protein tag to which specific, high avidity monoclonal antibodies have been developed. This tag is particularly useful in assays that require specific recognition by antibodies, such as immunoprecipitation. Cells were treated after transfection with thapsigargin to induce endoplasmic reticulum (ER) stress and subsequently treated with proteasome inhibitor MG132 before harvest. Cells were then lysed and immunoprecipitated using anti-FLAG M2 antibody with Protein G beads. Elution was then carried out, followed by a second immunoprecipitation with anti-HA antibody. The eluates were immunoblotted with the relevant antibodies.

## Cellular ubiquitination assay

To assay the ability of UBOX5 variants to ubiquitinate BIP, cells were transfected with MYC-tagged BIP, HA-tagged Ubiquitin, wildtype UBOX5, or UBOX5 variants. An empty vector served as negative control. All vectors included green fluorescent protein (GFP), allowing comparison of transfection efficiency between wildtype UBOX5 or variant UBOX5 by immunoblotting for GFP. Cells were then treated with thapsigargin after 24 h, harvested, lysed, and immunoprecipitated with antibodies against MYC. Elution was performed and eluates were subsequently immunoblotted with antibodies against HA and MYC.

## Pulse chase experiments

$7 \times 10^5$ HEK 293 cells were plated per wells of a 6 well plate. 24 hours later, 250 ng of empty vector or vector encoding wild-type UBOX5 were transfected into cells. At 24 hours post transfection, cells were then treated with 0.25 μM of Tharpsigargin (Sigma) for 2 h, and then 150 μM of cycloheximide (CST) was added at time = 0. Cells were then harvested at indicated times after cycloheximide induction for immunoblotting. The same procedure was followed for comparing wild-type UBOX5 with variant UBOX5 (D33N, K291R, R301Q, and S465C), and 165 ng of vector encoding wild-type or variant UBOX5 was transfected.

**Cellular fractionation experiments.** $3 \times 10^6$ HEK 293 cells were plated in 10 cm dish. HEK293 cells were transiently transfected with 1.8 μg wild-type UBOX5 expression plasmid per 10 cm dish and treated with 0.7 μM Thapsigargin (#sc-24017, Santa Cruz) for 16 h. Cells were then collected and lysed in Subcellular (SF) Buffer (250 mM Sucrose, 20 mM HEPES (pH 7.4), 10 mM KCl, 1.5 mM $MgCl_2$, 1 mM EDTA, 1 mM EGTA, 1 mM DTT, PPI cocktail), and left to rotate on a rotary shaker for 30 min at 4 °C. The resultant whole cell extracts were first centrifuged at 720 x g. The resultant nuclear pellet was then washed with the SF Buffer and re-centrifuged at 720 x g at 4 °C for 5 min, then lysed with RIPA and to generate the nuclear fraction. The post nuclear supernatant was then centrifuged at 10,000 x g for 10 min at 4 °C. The resultant supernatant (cytosolic and membrane fraction) was collected and subsequently ultracentrifuged at 100,000 x g for 1 h at 4 °C. The supernatant (this is the cytosolic fraction) was transferred to a fresh tube, while the pellet was resuspended and washed with the SF Buffer and ultracentrifuged at 100,000 x g for 1 h at 4 °C. The pellet was then lysed in RIPA (this is the endoplasmic reticulum (ER) fraction). The fractions were immunoblotted with antibodies against UBOX5 (#NBP1-

81469, Novus Biologicals), BIP (#3177, Cell Signaling Technology), Histone H2B (#2722, Cell Signaling Technology), Calnexin (#2679, Cell Signaling Technology), GAPDH (#sc-47724, Santa Cruz) and GFP (ab13970, Abcam).

### Antibodies used for biological experiments

Anti-c-Myc (9E10) (Santa Cruz, #sc-40, lot #C2224) was used for the immunoprecipitation (IP) pull-down of the UBOX5 and its variants. Anti-FLAG (M2) (Sigma Aldrich, #F1804, lot #SLCM4081) and anti-HA (Roche, #11666606001) were used for the tandem IP pull-down. Anti-HA (Proteintech, #51064-2-AP, lot #00061770), anti-c-Myc (A-14) (Santa Cruz, #sc-789, lot #C1314), anti-GRP78/BIP (Proteintech, #11587-1-AP, lot #00114059), anti-UBOX5 (Novus Biologicals, #NBP1-81469, lot #R30897), anti-GAPDH (0411) (Santa Cruz, #47724, lot #H2521), and anti-GFP (Abcam, #ab13970, lot #1018753-2) were used for immunoblotting (IB).

Custom anti-mouse UBOX5 (Genemed, lot #37002-37004) was used for immunohistochemistry (IHC), immunofluorescence (IF), and IB of mice sections and NIH-3T3 cell line.

### Reporting summary

Further information on research design is available in the Nature Portfolio Reporting Summary linked to this article.

## Data availability

The exome-wide summary statistics for the discovery whole exome sequencing analysis are appended in Supplementary Data 1. The discovery whole exome sequencing dataset in PLINK format have been deposited in the NGDC OMIX archive under accession ID OMIX007093. The dataset is available under controlled access due to the sensitive nature of individual level genotypes from whole-exome sequencing data, access can be obtained by requesting via the NGDC OMIX archive website (https://ngdc.cncb.ac.cn/omix/releaseList). Please contact Dr Li (liz11@gis.a-star.edu.sg) for data access. A response can be expected within 1 week. Source data are provided with this paper.

## Code availability

The following software were used; Burrow-Wheeler Aligner software (version 0.7.17-r1188) for mapping sequence reads (https://sourceforge.net/projects/bio-bwa/files/). Genome Analysis Tool Kit (version 4.1.3.0) for variant calling (https://github.com/broadinstitute/gatk/releases). VCFTOOLS (version 0.1.16 https://vcftools.github.io), BCFTOOLS (version 1.18, https://vcftools.github.io), and PLINK (v2.00) for quality control of genetic data (https://www.cog-genomics.org/plink/2.0/). The ENSEMBL VEP (variant effect predictor)(release 110, grch37) was used to annotate variants (https://ftp.ensembl.org/pub/release-110/).

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

## Acknowledgements

Supported by the Biomedical Research Council, Agency for Science, Technology and Research, Singapore (to Z.Li., C.C.K); by the Singapore Ministry of Health's National Medical Research Council under its STaR Investigator Award (MOH-STaR19nov-0001; to T.A.) and the National Research Foundation Singapore under its Open Fund - Large Collaborative Grant (MOH-OFLCG21jun-0003; to T.A., Z.W., and C.C.K.), administered by the Singapore Ministry of Health's National Medical Research Council; by Universiti Sains Malaysia Research University Individual grant 1001/PPSP/812152 (to A.T.L-S); by the NIHR Biomedical Research Centre at Moorfields Eye Hospital and the UCL Institute of Ophthalmology (to A.P.K), by the General Research Fund, Hong Kong (14102122 & 14105320 to C.C.T). A.P.K is supported by a UK Research and Innovation Future Leaders Fellowship, an Alcon Research Institute Young Investigator Award and a Lister Institute for Preventive Medicine Award. K.V.S is supported by Fight for Sight (UK) and the Desmond Foundation. A.S. is supported by Moorfields Eye Charity. M.B.M is supported by São Paulo Research Foundation - FAPESP (2018/20628-8) and National Council for Scientific and Technological Development - CNPq (307352/2023-4). J.C.Z. is supported by Velux Stiftung (grant 1860). H.A. is supported by a grant from the Higher Education Commission (HEC), Pakistan (grant number 20-17501/NRPU/R&D/HEC/2021). "For the purpose of open access, the author has applied a Creative Commons Attribution (CC BY) licence to any Author Accepted Manuscript version arising". We thank Dr. Rajkumar Dorajoo and Mr Sim Kar Seng for analytical support. S.A-O acknowledges support from King Saud University through the Vice Deanship of Scientific Research Chair and Glaucoma Research Chair in Ophthalmology

## Author contributions

Z.Li. analyzed the discovery stage whole-exome sequencing data. Z.Liu and W.L.C. performed all functional biological assays on UBOX5, together with N.N.M., Y.L., D.Z.J.T., P.M.X.Y., and W.W.L.S. T.D., M.N., L.J.C., F.T., M.E.N., M.O., S.N., T.K., S.A.P., R.H., T.T.L.W., C-Y.C., C.L.H., K.A-A., H-T.W., M.B.dM., N.D.T.N.H., N.V.T., N.T.T.H., Y.A., R.P-G., P.P.M.C., E.A., D.G., P.N., E.P., A.L., A.C., V.K., H.A., S.M., Y.Y.A., E.U.L., A.F., P-Y.B., C.C.Khaing, Y.M.A., R.D.R., E.S.P., D.G.M., I.C.V., G.P., Y.I., Y.T., M.T., N.O., M.U., J.P.C.d.V., V.P.C., R.Y.A., B.B.d.S., G.B.F., V.V.C., R.F., M.L.G-F., F.A., M.A., M.A.T.C., I.R.F., W.Y.M., C.S.L., C.L., H.M.T., K.T.O., P.P.S-K., P.F., F.P., R.Q., Z.M-D., N.L.O., L.D., S.H.S., V.H.D., R.Q., J.M.N., S.A-O., C.C.T., K.M., C.S., S.K., A.G.K., A.T.L-S., J.C.Z., N.H.D., P.F., K.T., C.P.P., A.P.K., and T.A. coordinated and managed sample collection / determination of PACG phenotypes. Z.X. contributed to Sanger capillary sequencing analysis of UBOX5 variants. X.Y.C and Y.T.L. performed whole-exome capture and built sequencing libraries in the laboratory. A.A., X.B., and Y.J.K. performed co-IP coupled to Mass Spectrometry to identify binding partners to UBOX5. R.C., A.S.Y.C., Y.S.L., and C.E.H.H. performed histological work on UBOX5 in mouse eye tissues. K.V.S. M.I.B., A.S., performed association analysis in UKBB. Z.W. led the effort to investigate the biology of UBOX5, including experimental design. B.C. contributed sequencing resources and commented on the manuscript. C.C.K. jointly led the research effort together with Z.W., T.A., A.P.K., C.P.P., K.T., Z.Li, Z.Liu, W.L.C., T.D., M.N., and L.J.C. C.C.Khor. wrote the manuscript, with inputs from co-authors.

## Competing interests

A.P.K has acted as a paid consultant or lecturer to Abbvie, Aerie, Allergan, Google Health, Heidelberg Engineering, Novartis, Reichert, Santen, Thea and Topcon. The remaining authors declare no competing interests.

## Additional information

Zheng Li [1,47], Wee Ling Chng[1,2,47], Zhehao Liu[1,2,47], Tan Do[3,47], Masakazu Nakano [4,47], Li Jia Chen [5,47], Yunhua Loo[2,6], Anita S. Y. Chan [2,6], Fotis Topouzis[7], Monisha E. Nongpiur[2,6], Mineo Ozaki[2,6,8], Satoko Nakano[9], Toshiaki Kubota [9], Shamira A. Perera[2,6], Rahat Husain[2,6], Tina T. L. Wong[2,6], Ching-Yu Cheng [2,6], Ching Lin Ho[2,6], Khaled Abu-Amero[10], Hon-Tym Wong[11], Mônica Barbosa de Melo[12], Nguyen Do Thi Ngoc Hien[3], Nguyen Van Trinh[3], Nguyen Thi Thanh Huong[3], Yaakub Azhany[13], Rodolfo Perez-Grossmann[14], Poemen PM Chan [5], Kelsey V. Stuart[15], Mahantesh I. Biradar [15], Anita Szabo[15], Eleftherios Anastasopoulos[7], Dimitrios A. Giannoulis[7], Panagiota Ntonti [7], Evangelia Papakonstantinou[7], Alexandros Lambropoulos [16], Anthoula Chatzikyriakidou[16], Vassilis Kilintzis[7], Humaira Ayub[17], Shazia Micheal[18], Yee Yee Aung[19], Edgar U. Leuenberger[20,21], Antonio Fea [22], Naing Naing Mon[2], Amihan Anajao[23], Xuezhi Bi [2,23], Yee Jiun Kok [23], Rachel S. Chong[2,6], Pui-Yi Boey[2,6], Darrell Zi Jing Tan[1], Wendy Wan Ling Sin[2], Balram Chowbay[24], Chaw Chaw Khaing[25], Yin Mon Aung[26,27], Rigo Daniel Reyes[28], Evangelia S. Panagiotou[7], Dimitrios G. Mikropoulos[7], Irini C. Voudouragkaki[7], Georgios D. Panos [7,29], Zhicheng Xie[1], Xiao Yin Chen[1], Yi Ting Lim[1], Wee Yang Meah[1], Ying Shi Lee[6], Candice Ee Hua Ho[6], Pearlyn Mei Xin Yeo[2], Yoko Ikeda[30], Yuichi Tokuda [4], Masami Tanaka[4], Natsue Omi[4], Morio Ueno [30], José P. C. de Vasconcellos[31], Vital P. Costa[31], Ricardo Y. Abe[31], Bruno B. de Souza[12], Guillermo B. Fong[32], Vania V. Castro[33], Ricardo Fujita[34], Maria L. Guevara-Fujita[34], Farah Akhtar[35], Mahmood Ali[35], Mary Ann T. Catacutan[20], Irene R. Felarca[20], Chona S. Liao[20], Carlo Lavia[22], Hlaing May Than[36], Khin Thida Oo[26], Phyu P. Soe-Kyaw[26], Paolo Frezzotti[37], Francesca Pasutto [38], Raquel Quino[28,39], Zaw Minn-Din[26], Nay Lin Oo[25], Laura Dallorto[22], Saw Htoo Set[19], Vi Huyen Doan[40], Raheel Qamar[41,42], Jamil Miguel Neto [31], Saleh Al-Obeidan [43], Clement C. Tham [5], Kazuhiko Mori [30], Chie Sotozono[30], Shigeru Kinoshita [44], Anastasios G. Konstas[7], Ahmad Tajudin Liza-Sharmini [13], Juan C. Zenteno[45,46], Nhu Hon Do[3], Paul J. Foster [15], Kei Tashiro[4,48], Chi Pui Pang[5,48], Anthony P. Khawaja [15,48], Tin Aung[2,6,48], Zhenxun Wang [1,2,6,48] ✉ & Chiea Chuen Khor [1,48] ✉

[1]Genome Institute of Singapore (GIS), Agency for Science, Technology and Research (A*STAR), 60 Biopolis Street, Genome #02-01, Singapore 138672, Republic of Singapore. [2]Duke-NUS Medical School, National University of, Singapore, Singapore. [3]Vietnam National Institute of Ophthalmology, Hanoi, Vietnam. [4]Department of Genomic Medical Sciences, Kyoto Prefectural University of Medicine, Kyoto, Japan. [5]Department of Ophthalmology and Visual Sciences, The Chinese University of Hong Kong, Hong Kong SAR, China. [6]Singapore Eye Research Institute, Singapore National Eye Centre, Singapore, Singapore. [7]1st Department of Ophthalmology, School of Medicine, Aristotle University of Thessaloniki, Thessaloniki, Greece. [8]Ozaki Eye Hospital, Hyuga, and Department of Ophthalmology, Faculty of Medicine, University of Miyazaki, Miyazaki, Japan. [9]Department of Ophthalmology, Oita University Faculty of Medicine, Yufu, Japan. [10]Department of Research, King Khaled Eye Specialist Hospital and Research Center, Riyadh, Saudi Arabia. [11]Dept of Ophthalmology, Tan Tock Seng Hospital, NHG Eye Institute, Singapore, Singapore. [12]Centro de Biologia Molecular e Engenharia Genética, Universidade Estadual de Campinas, Campinas, SP, Brazil. [13]Department of Ophthalmology and Visual Science, School of Medical Sciences, Health campus, Universiti Sains Malaysia and Hospital Pakar Universiti Sains Malaysia, Kubang Kerian, Kelantan, Malaysia. [14]Instituto de Glaucoma y Catarata, Lima, Perú. [15]NIHR Biomedical Research Centre, Moorfields Eye Hospital NHS Foundation Trust & UCL Institute of Ophthalmology, London, UK. [16]Laboratory of Medical Biology - Genetics, Medical School, Aristotle University of Thessaloniki, Thessaloniki, Greece. [17]Department of Biological and Health Sciences, Pak-Austria Fach-hochschule: Institute of Applied Sciences and Technology, Mang, Haripur, KPK, Pakistan. [18]Department of Ophthalmology, Radboud University Medical Centre, Nijmegen, the Netherlands. [19]Mandalay Eye department, Mandalay Eye ENT hospital, University of Medicine Mandalay, Mandalay, Myanmar. [20]Asian Eye Institute, Manila, Philippines. [21]University of the East Ramon Magsaysay College of Medicine, Manila, Philippines. [22]Dipartimento di Scienze Chirurgiche - Universita' di Torino, Turin, Italy. [23]Bioprocessing Technology Institute (BTI), Agency for Science, Technology and Research (A*STAR), 20 Biopolis Way, #06-01 Centros, Singapore, Singapore. [24]National Cancer Centre of, Singapore, Singapore. [25]Department of Ophthalmology, No(1) Defence Services General Hospital, Yangon, Myanmar. [26]Myanmar Eye Centre, Grand Hantha International Hospital, Yangon, Myanmar. [27]University Hospitals Coventry and War-wickshire Trust, Coventry, United Kingdom. [28]Department of Ophthalmology/Glaucoma Section, Asian Hospital & Medical Center, Muntinlupa City, Philippines. [29]Division of Ophthalmology and Visual Sciences, School of Medicine, University of Nottingham, Nottingham, UK. [30]Department of Ophthalmology, Kyoto Prefectural University of Medicine, Kyoto, Japan. [31]Departamento de Oftalmologia, Faculdade de Ciências Médicas, Universidade Estadual de Campinas, Campinas, SP, Brazil. [32]Instituto de Ciencias Medicas, Lima, Perú. [33]Hospital Nacional Arzobispo Loayza, Lima, Perú. [34]Centro de Genetica y Biologia Molecular, Universidad de San Martin de Porres, Lima, Perú. [35]Department of Glaucoma, Al-Shifa Trust Eye Hospital, Rawalpindi, Pakistan. [36]Department of Ophthalmology, North Okkalarpa General Hospital, Yangon, Myanmar. [37]Surgery Department, Section of Ophthalmology, University of Siena, Siena, Italy. [38]Humangenetisches Institut, Universitätsklinikum Erlangen, FAU Erlangen-Nürnberg, Kussmaulallee 4, 91054 Erlangen, Germany. [39]University of the Philippines- College of Medicine, Manila, Philippines. [40]Hue University of Medicine and Pharmacy, Danang City, Vietnam. [41]Science and Environment Sector, ICESCO, Rabat, Morocco. [42]Pakistan Academy of Sciences, Islamabad, Pakistan. [43]Department of Ophthalmology, College of Medicine, King Saud University, Riyadh, Saudi Arabia. [44]Department of Frontier Medical Science and Technology for Ophthalmology, Kyoto Prefectural University of Medicine, Kyoto, Japan. [45]Rare Disease Diagnostic Unit/Biochemistry Department, Faculty of Medicine, UNAM, Mexico City, Mexico. [46]Genetics Department, Institute of Ophthalmology "Conde de Valenciana", Mexico City, Mexico. [47]These authors contributed equally: Zheng Li, Wee Ling Chng, Zhehao Liu, Tan Do, Masakazu Nakano, and Li Jia Chen. [48]These authors jointly supervised this work: Kei Tashiro,Chi Pui Pang,Anthony P. Khawaja,Tin Aung,Zhenxun Wang and Chiea Chuen Khor. ✉e-mail: zhenxun.wang@duke-nus.edu.sg; khorcc@a-star.edu.sg

