## [Transparent Peer Review file · Nature Communications]

Functionally deficient UBOX5 variants and primary angle-closure glaucoma

Corresponding Author: Professor Chiea Chuen Khor

Version 0:

Reviewer comments:

Reviewer #1

(Remarks to the Author)

Li and colleagues described genetic associations of rare variants in the UBOX5 gene with primary angle closure glaucoma (PACG), which is one of the leading causes of irreversible blindness worldwide. Using whole exome sequencing (WES), the authors narrowed down to a few chosen pathogenic variants within this gene for replication across multiple populations. Functional analysis based on the expressions of UBOX5 in appropriate ocular tissues and its physical interactions with downstream targets suggested its potential involvement in PACG. The functionally associated variants were further validated in two additional populations. PACG and its endophenotypes such as intraocular pressure (IOP) could have strong hereditary components comprising minor involvements of rare variants (with Mendelian influences) along with several common variants with polygenic influences. The authors have focused primarily on the rare and pathogenic coding variants and have implicated UBOX5 variants with an increased risk of PACG. The manuscript is reasonably well written, and the methodology, although needing some justifications, is generally sound. The results would need additional clarifications, and the discussion is too concise and would require elaboration.

The primary strengths of this study include the discovery of associations of UBOX5 variants with PACG in sufficiently large populations and replications of the same across multiple populations worldwide that strengthens the possibility of genetic predisposition. Modest functional assessments supported the genetic association and suggested the likely involvement of UBOX5 in disease pathogenesis. However, the general weaknesses (some of which are also acknowledged in the limitations) include (and not limited to) the lack of stringent clinical definitions for cases and controls and non-uniform methods of enrolment across populations; relatively less rigorous assessments of associated variants in a complex disease like PACG; many insights on the involvement of UBOX5 that are built from nebulous connections; and lack of mechanistic details in implicating the actual function(s) of UBOX5 in PACG pathogenesis.

My major concerns are as follows:

1) Some of the sentences in the abstract need to be rephrased. Statements like “Significantly associated genes”, “Carriers of rare, protein-altering variants at UBOX5” and “functionally deficient variants were enriched in PACG cases” are vague unless indicated by actual numbers (n) and/or their percentages. The final statement should be toned down significantly as the authors have only demonstrated a functional association of the UBOX5 – BIP signalling pathway and that does not necessarily indicate its direct impact on disease pathogenesis.

2) The hypothesis lacks clarity, and the rationale needs to be clear. The authors should enumerate the advantages of using WES over and above the efforts of earlier genome wide association studies (GWAS). It is unclear if screening was restricted to the coding regions only or were the promoters and splice variants covered? If so, how much of the exonic regions were padded to include the splice sites? Without the screening of the splice sites and the regulatory regions of the promoters, the functional assessments of the associated gene(s) and their downstream targets would be grossly inadequate.

3) The clinical criteria for enrolment of cases and controls need to be stringent and harmonized across all the populations. Unfortunately, other than the cohorts enrolled from Singapore and Hong Kong, the enrolment criteria for all the remaining populations are too vague and lack specifics. As PACG is an age-related disease, it would be useful to expand supplementary table 1 and include the mean ages of the subjects along with the distributions of gender across these populations. Apparently, some of the populations are ethnically diverse, but their demographic details are elusive. Since it is

an association study, how were the cases and controls matched to age, gender and ethnicity within these populations?

4) There is no discussion of caveats pertaining to clinical presentations and risk factors. It appears that all cases of acute primary angle closure (APAC) and PACG were pooled together. What proportions of cases underwent laser peripheral iridectomy with resolved angle closures? Were they excluded from final analysis? Globally, at least half of the APAC patients would progress to PACG with elevated IOP. If all categories of cases were pooled, it would be better to term this as "primary angle closure disease (PACD)" rather than PACG. Since carriers of UBOX5 variants had significantly elevated risk of APAC along with greater risk in females, it would be useful to provide the gender data across the clinical subtypes. I assume there was no skewing of gender data across these populations.

5) The classifications of pathogenic variants need justification. The authors should use more rigorous tools like the REVEL scores rather than Polyphen2. Since REVEL scores encompasses 13 different tools (including Polyphen2), it is more reliable in terms of predictions for missense variants. I would suggest using REVEL scores for reclassifying the observed variants.

6) While the data was consistent with regards to the UBOX5 qualifying variants across all the discovery populations, were there variant(s) of other associated gene(s) that were overrepresented amongst these cases?

7) It would also be useful to know if the carriers of UBOX5 qualifying variants also harbored the GWAS associated variants? Did the presence of an associated common variant along with the UBOX5 rare variant increase the susceptibility to APAC and/or PACG? Likewise, did the presence (or absence) of additional variants of other associated genes along with UBOX5 variants exhibit enhanced risk of the disease? Clinically, did the presence of multiple alleles (rare and common) impact disease severity in terms of prognosis? How were the carriers of UBOX5 qualifying variants different in terms of their disease manifestations compared to those who did not harbor these variants?

8) Since the variants are rare and present in lesser numbers across populations, it is helpful to get an overall estimate of their burden by adding them (as was done by the authors). But biologically, the types of these variants would exhibit exclusive cellular and molecular functions based on their protein domains and specific interactions. This possibility needs to be discussed in detail.

9) It is unlikely that variants in UBOX5 would be solely responsible for PACG. A complete profile of the downstream targets of UBOX5 should be provided through a network analysis. This would also demonstrate the potential functional interactions of other associated genes (both obtained from WES and those identified from earlier GWAS) along with UBOX5.

10) The functional basis of UBOX5 being directly involved in PACG is somewhat "over-stated" and much of the discussion is speculative. There are repeated statements inferring that a specific function for UBOX5 has been identified – which was not. I would have expected more mechanistic details from this consortium as additional downstream targets could have been analysed under a specific model system. Since PACG involves structural defects and IOP changes may occur independently of these defects, it would have been worthwhile to assay the interactions of UBOX5 and BIP and other targets in presence and absence of elevated IOP. This would have also implied if structural abnormalities had complex (or no) relationships to IOP. Localizations of UBOX5 in the relevant ocular tissues is only suggestive of its involvement. But unless experiments are designed to demonstrate that perturbations of its normal functions and interactions lead to PACG or its the endophenotypes, and that these are reversed through appropriate rescue mechanisms, its causal role in PACG pathogenesis would remain elusive.

Some minor concerns pertain to few typos and syntax errors and use of "95%CI" and not "95%ci" throughout the text.

Reviewer #2

(Remarks to the Author)

The manuscript presents a well-designed and comprehensive study that investigates the role of UBOX5 variants in Primary Angle-Closure Glaucoma (PACG). By combining whole exome sequencing (WES), replication studies, and functional assays, the study provides novel insights into the genetic underpinnings of PACG and highlights UBOX5 as a gene of interest. The strengths of the study include its rigorous methodology, large sample sizes, and the functional validation of UBOX5 variants, which add substantial value to the genetic association findings. However, there are areas where the manuscript could be strengthened, particularly in terms of mechanistic insights and clarity on variant selection. Below are specific comments and suggestions for improvement.

Major Comments

1. The authors have discussed the limitations in the manuscript text, which is reflected in the Figure 1 where in Post-hoc analyses on sex stratification and risk of angle-closure glaucoma except the Japanese cohort all other cohorts are not showing significant risk of having the diseases especially in men from all the four sites.
2. While the study identifies UBOX5 variants as significant contributors to PACG, the mechanistic link between UBOX5's ubiquitination of BIP and PACG development is not fully explored.
3. The major criticism of the manuscript lies upon the choice of a generic cell line like HEK293 and overexpression of proteins using transient transfection. The nature of intricate cellular and biochemical analyses depicted in this manuscript requires finer experiments with stable cell lines and validation in an in-vivo system.
4. In Figure 2A, FLAG is used for IP but no IB: FLAG is shown. IB:HA should also be included here to shed more light on ubiquitinated BIP. In the bottom panel lane 2, it is not clear what are lower bands in the 1st IP input lane.

5. In Figure 2B, are myc-tagged BIP, UBOX5 and HA-tagged ubiquitin in same plasmid backbone? If yes, is it the only empty vector that has been taken as control? If not, where are other empty vector controls for this experiment? Where are the controls for Thapsigargin and MG132? Those results should also be included here. Also How did the authors tackle of co-transfecting three different constructs transiently and making sure that the cells are receiving all three constructs. Besides, there could be payload issue too.
6. In Figure 3A, gel quantitation data is required for the immunoblot.
7. In Figure 3B, the authors should show the status of endogenous UBOX5 in HEK293 cells. In addition, the authors should also include the nuclear fraction data immunoblotted with UBOX5, BIP, GAPDH and calnexin to establish the specificity of their fractionation method especially in the context of ER and nuclear fraction. They might choose to use LaminA/C as a marker for the nuclear fraction.
8. In Figure 3C, where is the empty vector control in the immunoblot analyses?
9. The experiment depicted in Figure 4, does not necessarily prove UBOX5 is solely responsible for the physical interaction between BIP and ubiquitin. The authors should first perform an experiment by in-vitro pull-down to detect the domain in UBOX5 is responsible for binding/deposit of ubiquitin on BIP. Afterwards another experiment is required upon making deletion constructs of UBOX5 to prove that really perturbs the binding/deposit of ubiquitin on BIP. Also, gel quantitation data with number of biological replicates with proper statistical analyses should also be included.
10. In Table 1, since most of these mutations are rare in populations, CADD score is not enough. Ideally these should be tested using various bioinformatics tools such as SIFT, LRT, MutationTaster, Provean etc. and it should be considered deleterious when at least three of these tools designate the mutation as deleterious. The authors could consider including a matrix showing how their functional annotation is corroborating with the bioinformatics prediction tools and provide a graph where functionally deficient mutations will be assessed further with their prediction scores generated using different tools.
11. Further, the manuscript would benefit from additional experiments or a more detailed discussion on how this interaction could lead to the pathophysiological changes observed in PACG. Specifically, how does UBOX5-mediated mono-ubiquitination of BIP influence intraocular pressure, optic nerve damage, or other features of PACG? More clarity on this could significantly enhance the impact of the findings.
12. The manuscript highlights certain UBOX5 variants for functional analysis but does not provide a clear rationale for their selection. It would be helpful to include more information on the criteria used to prioritize these variants over others. Were these variants chosen based on their frequency, predicted pathogenicity, or previous associations with PACG? Providing this context would improve the transparency of the study design and help readers understand the significance of the chosen variants.
13. Although the authors address population stratification, the manuscript could benefit from a more detailed discussion of how genetic or environmental differences between the studied populations might influence the findings. For example, are there specific genetic factors in the Asian cohorts that could explain the stronger associations observed in these populations? Additionally, discussing potential limitations related to population differences in the replication cohorts would add depth to the analysis.

Minor Comments

14. The statistical genomics methods used are robust, and the use of multiple cohorts enhances the study's power. However, the manuscript would benefit from a clearer explanation of how the p-value threshold for significance was determined (e.g., why $P < 2.5 \times 10^{-6}$ was chosen for exome-wide significance). This would help readers better understand the stringency of the analysis.
15. The identification of UBOX5 and its impact on BIP ubiquitination presents potential therapeutic targets for PACG. How do the authors envision translating these findings into targeted therapies?
16. The manuscript is generally well-written, but there are a few areas where clarity could be improved. For example, the description of the immunoprecipitation assays in the methods section could be simplified to make it more accessible to readers who may not be specialists in this technique. Additionally, some sentences in the discussion section are quite long and could be broken down for easier comprehension.

Reviewer #3

(Remarks to the Author)

Association between PACG and rare, protein-altering genetic variants were found in UBOX5 through exome sequencing of thousands of patients and controls. Results were replicated in an additional study also of thousands of patients and controls. UBOX5 is expressed in the iris sphincter pupillae, optic nerve head, and throughout the retina. Co-IP and mass spec revealed that three heat shock proteins interacted with UBOX5. One of these, Binding immunoglobulin protein (BIP) was prioritized in particular due to its role as an inducer of the unfolded protein response pathway which are linked to glaucoma. Functional studies were done of 35 UBOX5 variants found to be enriched in PACG cases. 21/35 were found to be functionally impaired, and were carried more often in people with PACG versus controls. These functional studies were also replicated in additional cases. The authors suggest that the UBOX5-BIP pathway is involved in PACG pathogenesis.

Comments:

Authors are aware that the case control study design does not allow assignment of causal relationships. They are also aware of potential issues with determination of PACG status based upon diagnosis codes in the UK biobank rather than in a hospital. Studies were replicated at every stage, however, and this reviewer has confidence in the validity of both the genetic and biochemical analyses.

It is not really clear to me how these genes are linked to PACG, despite the authors assertion that UBOX5-BIP pathway is involved in glaucoma pathogenesis. The authors speculate that mono-ubiquitination of BIP by UBOX5 could aid in protecting retinal ganglion cells from ER stress and glaucomatous optic neuropathy, and functionally deficient UBOX5 might

remove this layer of protection, but it is not clear to me how, or indeed if this occurs. Additional discussion and/or exploration of this connection would be useful.

Version 1:

Reviewer comments:

Reviewer #1

(Remarks to the Author)

The authors have addressed to all my concerns and have provided appropriate evidences and justifications.

Reviewer #2

(Remarks to the Author)

This reviewer appreciates the way authors have addressed all the points raised by this reviewer. However, a few points need more clarification.

1. For point#2, it would be great if the authors could show the same blots without the pulse of Thapsigargin.
2. Also, while checking on the effect of UBOX5 mutants on BIP, it has been observed that the overall expression of R301Q mutant is low. In that case, how could the authors confirm that the effect of R301Q mutant on BIP is not due to its low abundance. It is important to comment on that, since the other mutant D33N, which should not have any effect on BIP (the authors pointed this out), varies proportionately with D33N abundance for last two time points.
3. This reviewer does not quite agree with the authors on the response for point#3. BIP might be ubiquitously expressed in all cell types, however, there are multiple reports that suggest expression of BIP varies not only between different cell types but under different conditions such as cell stress and diseases states.

Overall, this reviewer feels that while these biochemical assays using a recombinant protein in a generic cell line delve into gross mechanistic underpinnings at cellular level, however, it still lacks correlation with specific disease phenotype/trait, knowing which could aid in further therapeutic interventions in a debilitating disease like angle closure glaucoma.

Reviewer #3

(Remarks to the Author)

The authors have addressed all the concerns I had with the previous version. This is an exciting paper that will be of considerable interest to researchers in the field.

Version 2:

Reviewer comments:

Reviewer #2

(Remarks to the Author)

This reviewer has no further comment. Just a clarification-the figure where bip expression is measured in different time points in wt UBOX5 and different mutational background, the D33N blot IB-Bip has vertical lines visible between time points. Are these from different blots clubbed together? The same lines are observed between lanes in D33N: IB-UBOX5, D33N: IB-GFP, K291R: IB-UBOX5, R301Q: IB-GFP.

We are very grateful to the three Referees for their helpful comments in the first round of review. We present our responses to their comments in this document.

Original comments from the Reviewers are in **black** font.

Our responses are in **blue** font.

Revisions to the manuscript are indicated in **red** font.

Figure legends for figures presented in this response document are in **green** font.

Reviewer #1 (Remarks to the Author):

Li and colleagues described genetic associations of rare variants in the UBOX5 gene with primary angle closure glaucoma (PACG), which is one of the leading causes of irreversible blindness worldwide. Using whole exome sequencing (WES), the authors narrowed down to a few chosen pathogenic variants within this gene for replication across multiple populations. Functional analysis based on the expressions of UBOX5 in appropriate ocular tissues and its physical interactions with downstream targets suggested its potential involvement in PACG. The functionally associated variants were further validated in two additional populations. PACG and its endophenotypes such as intraocular pressure (IOP) could have strong hereditary components comprising minor involvements of rare variants (with Mendelian influences) along with several common variants with polygenic influences. The authors have focused primarily on the rare and pathogenic coding variants and have implicated UBOX5 variants with an increased risk of PACG. The manuscript is reasonably well written, and the methodology, although needing some justifications, is generally sound. The results would need additional clarifications, and the discussion is too concise and would require elaboration.

Response: We agree with the Reviewer and have tried our best to provide clarifications on the specific points raised. Also, the Discussion section is now elaborated upon and thoroughly revised.

The primary strengths of this study include the discovery of associations of UBOX5 variants with PACG in sufficiently large populations and replications of the same across multiple populations worldwide that strengthens the possibility of genetic predisposition. Modest functional assessments supported the genetic association and suggested the likely involvement of UBOX5 in disease pathogenesis.

However, the general weaknesses (some of which are also acknowledged in the limitations) include (and not limited to) the lack of stringent clinical definitions for cases and controls and non-uniform methods of enrolment across populations;

Response: The clinical definitions that we applied is in keeping with the most recent case-definitions of PACG (see for example <https://www.rcophth.ac.uk/wp-content/uploads/2021/10/The-Management-of-Angle-Closure-Glaucoma-Clinical-Guidelines.pdf>). The unifying characteristic of patients with APAC (acute primary angle-closure) and PACG (primary angle closure glaucoma) is that the cause of their angle closure **was not secondary to any other pathology** (hence classified as primary angle

closure), and that all patients have gonioscopic angle closure with significant glaucoma disease severity.

Because not all the case-control collections reported in our manuscript have patients with the APAC sub-classification, primary analysis was focused on the overall PACG phenotype, which included both acute and chronic forms. The **Methods** section of the main text has been revised to make this explicit.

relatively less rigorous assessments of associated variants in a complex disease like PACG;

Response: We beg to differ that the associated variants underwent 'relatively less rigorous assessments.' This was because for primary analysis using the gene-based burden test, the selection of qualifying variants was uniformly applied across all study samples (all variants with CADD score >10; such a score reflects predicts these variants to be within the top 10% most deleterious substitutions in the human genome).

The consistency of the association between a higher burden of *UBOX5* qualifying variants in patients with PACG compared to controls across all 14 independent studies of different ancestries suggest the statistical assessment to be rigorous.

many insights on the involvement of *UBOX5* that are built from nebulous connections; and lack of mechanistic details in implicating the actual function(s) of *UBOX5* in PACG pathogenesis.

Response: We agree with the Reviewer and have refrained from making strong statements regarding the actual function of *UBOX5* in PACG pathogenesis. After establishing statistically significant association between carriage of *UBOX5* protein-altering variants and increased susceptibility to PACG, we pursued laboratory experiments to elucidate the biological function of *UBOX5*, beginning with substrate trapping approaches by performing immunoprecipitation of a *UBOX5*-ubiquitin binding domain (UBD) fusion protein to find the actual targets of *UBOX5*.

The substrate trapping approach (Watanabe M et al., *Commun Biol* 2020 Oct 20;3(1):592; Mark KG et al., *Mol Cell* 2014; 53(1):148-61; Mark KG et al., *Nat Protoc* 2016; 11(2):291-301), which was developed to bypass the many limitations of in-vitro ubiquitination assays, showed that *UBOX5* potentially ubiquitinates three heat shock proteins (HSPA1B, HSPA8, and HSPA5) critical for maintaining cellular proteostasis. HSPA5 (also known as BIP) presented a very enticing target for further study, as it is both a sensor of endoplasmic reticulum (ER) stress and an inducer of the unfolded protein response (UPR)(Fribley A et al., *Methods Mol Biol.* 2009;559:191–204). We demonstrate **a)** interaction of BIP with *UBOX5* (**Figure 2A**) and **b)** ubiquitination of BIP by *UBOX5* using immunoprecipitation approaches (**Figure 2B**). We go on to show that most of the variants that are enriched in PACG cases compared to controls had less ubiquitination activity (**Table 1**, **Figure 4** and **Supplementary Figure 7**). Consistent with the notion that *UBOX5* ubiquitinates BIP, our cell fractionization experiments

demonstrated that UBOX5 was found in the ER (**Figure 3B**) where BIP normally resides.

Now in this revised manuscript, we additionally show that the presence of wild-type UBOX5 increases the half-life of BIP under ER stress conditions. Conversely, co-expression of functionally deficient UBOX5 alleles, including point mutants within the important UBOX domain of UBOX5, did not increase the half-life of BIP. This is an important result because BIP was recently shown to be short-lived (Shim SM et al., *Sci Signal* 2018 Jan 2;11(511):eaan0630) and there is a possibility that ubiquitination of BiP might lead to increased turnover, in turn contradicting our hypothesis. Because BIP has been well-described to have cytoprotective properties in responding to ER stress (Kudo T et al., *Cell Death Differ.* 2008; 15(2):364-75; James AW et al., *Neurochem Int* 2023 Oct;169:105573; Morris JA et al., *J Biol Chem.* 1997; 272(7):4327-34; Wang J et al., *eLife.* 2014 Jul 22; 3:e03496), and has a key role in protein quality control within the cell (Chen X et al., *Signal Transduct Target Ther* 2023; 8(1):352), the observation that UBOX5 increases the half-life of BIP provides additional support that UBOX5 – BIP may have cytoprotective properties. BIP is induced during ER-stress to increase protein folding capacity of the cell. Increased half-life under these conditions would enable cells to further boost their proteostatic capacity, enabling more efficient resolution of ER stress and return to homeostasis.

We have now revised the manuscript by adding these sentences into the discussion section: “Firstly, the ubiquitination of BIP by UBOX5 appeared to increase the half-life of BIP. In agreement with prior reports (Shim SM et al., *Sci Signal* 2018 Jan 2;11(511):eaan0630) that BIP has a short half-life, we find that co-expression of UBOX5 led to decreased BIP protein turnover. BIP senses and responds to ER stress by inducing the unfolded protein response pathway. This pathway is conserved in mammalian cells (Fribley A et al *Methods Mol Biol.* 2009; 559:191–204) and in plants (Liu Y et al., *Front Plant Sci.* 2022 Oct 7;13:1019414), and cells with pathway deficiencies were significantly less able to survive stressful stimuli, suggesting that activation of the unfolded protein response pathway in response to ER stress confers cytoprotective properties. Protein abundance of BIP is induced during ER stress to increase cellular protein folding capacity. Increasing its half-life would thus further increase cellular protein folding capability under these conditions, and lead to a rapid resolution of ER stress and return to homeostasis.”

My major concerns are as follows:

- 1) Some of the sentences in the abstract need to be rephrased. Statements like “Significantly associated genes”, “Carriers of rare, protein-altering variants at UBOX5” and “functionally deficient variants were enriched in PACG cases” are vague unless indicated by actual numbers (n) and/or their percentages. The final statement should be toned down significantly as the authors have only demonstrated a functional association

of the UBOX5 – BIP signalling pathway and that does not necessarily indicate its direct impact on disease pathogenesis.

Response: We agree with the Reviewer. The abstract has been revised as follows to clarify:

a) The phrase “significantly associated genes” has been revised to: “Genes surpassing exome-wide significance ($P < 2.5 \times 10^{-6}$)”

b) The phrase “Carriers of rare, protein-altering variants at UBOX5” has been revised to: “Carriers of rare, protein-altering variants at *UBOX5* (observed in 154 out of 7,186 affected individuals [2.1%] and in 3,975 out of 477,197 unaffected controls [0.83%])...” to reflect actual numbers (n) and their percentages.

c) The phrase “functionally deficient variants were enriched in PACG cases” has been revised to: “Evaluation of the functional status of 35 UBOX5 variants using laboratory assays suggest that functionally deficient variants were enriched in PACG cases compared to controls. This was validated in an independent collection where 3 persons carrying functionally deficient variants were observed out of 208 cases (1.4%), whereas none were observed in 600 controls”.

D) The final sentence in the abstract has been toned down to read: “Our findings suggest the UBOX5 – BIP signalling pathway might be involved in PACG biology.”

2) The hypothesis lacks clarity, and the rationale needs to be clear. The authors should enumerate the advantages of using WES over and above the efforts of earlier genome wide association studies (GWAS). It is unclear if screening was restricted to the coding regions only or were the promoters and splice variants covered? If so, how much of the exonic regions were padded to include the splice sites? Without the screening of the splice sites and the regulatory regions of the promoters, the functional assessments of the associated gene(s) and their downstream targets would be grossly inadequate.

Response: We have clarified the hypothesis and rationale for using whole exome sequencing as follows in the revised manuscript: “We hypothesize that there are genetic variants with strong effect on PACG risk residing in the protein-coding regions (exome) of the genome”.

We have also enumerated the advantages of using WES over and above the earlier GWAS studies in this revised sentence in the main text: “While genome-wide association studies have identified significantly associated loci, pinpointing the exact causal genes within these loci is challenging due to the analysis of non-coding variants. This limitation hinders understanding the biological mechanisms of PACG. On the other hand, whole exome sequencing enumerates variants in the protein-coding sequence, offering direct insights into disease biology.” Canonical splice sites are covered by the

exome capture kit, as previously also described in Li Z et al., *JAMA*. 2021; 325(8):753-764.

Promoters are not covered because it is **very challenging to define actual promoters accurately using uniform, bioinformatic criteria**. This is the reason promoters are not routinely assessed in exome sequencing studies, as is seen in the following published examples:

- a) Cirulli ET et al., Exome sequencing in amyotrophic lateral sclerosis identifies risk genes and pathways. *Science* 2015; 347:1436-41
- b) Kenna KP et al., NEK1 variants confer susceptibility to amyotrophic lateral sclerosis. *Nat Genet*. 2016 Sep;48(9):1037-42
- c) Do R et al., Exome sequencing identifies rare LDLR and APOA5 alleles conferring risk for myocardial infarction. *Nature* 2015; 518:102-6
- d) Flannick J et al., Exome sequencing of 20,791 cases of type 2 diabetes and 24,440 controls. *Nature* 2019; 570:71-76

3) The clinical criteria for enrolment of cases and controls need to be stringent and harmonized across all the populations. Unfortunately, other than the cohorts enrolled from Singapore and Hong Kong, the enrolment criteria for all the remaining populations are too vague and lack specifics. As PACG is an age-related disease, it would be useful to expand supplementary table 1 and include the mean ages of the subjects along with the distributions of gender across these populations. Apparently, some of the populations are ethnically diverse, but their demographic details are elusive. Since it is an association study, how were the cases and controls matched to age, gender and ethnicity within these populations?

Response: We clarify here that the clinical criteria for enrolment of PACG cases were stringent. Firstly, all PACG cases fulfilled either of the two inclusion criteria used to define PACG:

- a) Patients with previous acute primary angle-closure
- b) Patients with chronic PACG

Patients with acute-primary angle closure were defined by the presence of at least two of the following symptoms: ocular pain, nausea and / or vomiting, with an antecedent history of blurring of vision; a presenting IOP of more than 28 mmHg on Goldmann applanation tonometry; and the presence of at least three of the following signs: conjunctival injection, corneal edema, mid-dilated non-reactive pupil, and a shallow anterior chamber. Acute-primary angle closure is an ocular emergency because it causes a rapid increase in intraocular pressure due to outflow obstruction of aqueous humor. If left untreated, irreversible optic nerve damage occurs.

Patients diagnosed with chronic PACG had asymptomatic closure of the angle assessed by gonioscopy, accompanied by glaucomatous optic neuropathy (GON), defined as abnormally large optic disc excavation and loss of neuro-retinal rim tissue with a vertical cup-to-disc ratio greater than the 97.5 percentile of the population when examined with a 78D bio-microscopic lens. A GON diagnosis was confirmed by the presence of visual field loss (consistent with glaucoma as per International Society of Geographical and Epidemiological Ophthalmology criteria) detected with static automated white-on-white threshold perimetry (program 24-2 SITA, model 750, Humphrey Instruments, Dublin, Ca).

Patients younger than 50 years were excluded, as were patients with secondary forms of angle closure glaucoma such as neovascular glaucoma. Unaffected controls were enrolled from the same geographic area as the PACG cases and had self-reported ancestry similar to PACG cases.

The robustness of the study design was affirmed with the discovery of UBOX5 as a susceptibility gene for PACG, with no other gene showing results surpassing exome-wide significance (indicating that background, residual experimental noise was well controlled for, in similar vein to our previous exome sequencing study for exfoliation syndrome / glaucoma; **JAMA**. 2021; 325(8):753-764).

In addition, the association between carriage of rare UBOX5 variants and increased risk of PACG was observed even after the following steps:

1. Stratifying by sex.
2. Stratifying by acute angle closure glaucoma compared to non-acute angle closure glaucoma.
3. Scrutiny by validation in independently ascertained samples.

For the discovery exome sequencing study from Singapore, Hong Kong, Vietnam and Japan, PACG cases and controls were of matching ethnicities (Chinese ancestry for Singapore and Hong Kong, Vietnamese (majority from the Kinh ethnic group, see for example Khor CC et al., **Nature Genetics** 2011; 43:1139-41) ancestry for Vietnam, and Japanese ancestry for Japan. The optimal ancestry matching between cases and controls was confirmed with genetic principal component analysis (do refer to section on “**Ancestry principal component calculation for whole exome sequenced samples**” in the **Methods** section of the revised main text as well as **Supplementary Figure 8**).

The same design was adopted for the validation collections, and the overall consistency of the *UBOX5* association across all 14 studies of various ancestries and sample collection mandates suggest that the results are robust.

4) There is no discussion of caveats pertaining to clinical presentations and risk factors. It appears that all cases of acute primary angle closure (APAC) and PACG were pooled together. What proportions of cases underwent laser peripheral iridectomy with resolved angle closures? Were they excluded from final analysis?

Response: We did not pursue secondary analysis on patients who underwent laser peripheral iridectomy whose angle closures subsequently resolved as the study was not designed to evaluate surgical outcomes. Our study methodology did not pre-specify exclusion of these patients, as the aim of the primary analysis was to evaluate the relationship between protein-altering variants and overall PACG risk.

Globally, at least half of the APAC patients would progress to PACG with elevated IOP. If all categories of cases were pooled, it would be better to term this as “primary angle closure disease (PACD)” rather than PACG.

Response: We are grateful for this comment and clarify that our study aim was to investigate the genetic basis of PACG which is a severe phenotype associated with blindness and visual loss. We understand the Reviewer's comment about primary angle closure disease (PACD), but we would like to point out that PACD encompasses a spectrum of disease that includes primary angle closure suspects (PACS, which is often an asymptomatic phenotype), primary angle closure and PACG. We have previously investigated PACG GWAS hits in PACS and found that not all PACG GWAS hits overlap with PACS (Nongpiur ME et al, *Ophthalmology* 2018 May;125(5):664-670). Thus, there could be some differences in the genetics of these subtypes of PACD especially with the inclusion of PACS.

Since carriers of UBOX5 variants had significantly elevated risk of APAC along with greater risk in females, it would be useful to provide the gender data across the clinical subtypes. I assume there was no skewing of gender data across these populations.

Response: In the discovery exome-sequencing study comprising 4,667 PACG patients and 5,473 controls, there were 2,894 PACG patients (62%) and 2,912 controls (53.2%) who were female. This was in keeping with previous observations reporting a higher incidence of PACG in females compared to males (Zhang N et al., *Front Med* (Lausanne) 2021 Jan 18;7:624179).

On stratifying the UBOX5 association with PACG according to sex (shown in **Figure 1** of the manuscript and appended in the following page for ease of reference), female carriers of *UBOX5* qualifying variants had nominally higher odds of PACG (OR=2.23, 95%ci: 1.53-3.25) compared to male carriers (OR=1.66, 95%ci: 0.96-2.89; **Figure 1a**). However, this sex-stratified difference in odds ratios between males and females was not statistically significant.

UBOX5 is associated with increased risk of primary angle-closure glaucoma (PACG) in the discovery whole exome sequencing study. For each study, the size of the square of the

odds ratio is proportional to the sample size. The 95% confidence intervals are reflected by the width of the horizontal line across the squares. Both primary analysis (on all participants) and post-hoc sex-stratified analysis for each collection are shown.

Moving to the 1,442 participants with acute angle-closure glaucoma, we observed that 1024 (71%) were female, also consistent with previous observations of more females having acute angle-closure glaucoma compared to males (Seah SK et al. *Arch Ophthalmol* 1997; 115:1436-40; Liu L et al., *Sci Rep* 2017; 7(1):14885, as well as Mehta SK et al., *Clin Ophthalmol* 2022; 16:2341-2351). Carriers of *UBOX5* qualifying variants had significantly elevated risk of acute primary angle closure subtype (OR=2.39, 95%ci, 1.56 – 3.66), with female carriers having 3-fold increased odds (95%ci: 1.81-4.97) compared to female non-carriers. We did not observe a significant association between male carriers of *UBOX5* variants and risk of acute angle-closure glaucoma, likely due to the small number of case patients (2 carriers out of 418). These results are shown on the following page for ease of reference, as well as in **Figure 1** of the manuscript main text.

Sex stratified analysis of *UBOX5* qualifying variants and risk of acute angle-closure glaucoma in the discovery whole exome sequencing study. For each study, the size of the

square of the odds ratio is proportional to the sample size. The 95% confidence intervals are reflected by the width of the horizontal line across the squares.

From the data, gender skewing should not be an issue because the analyses were specifically stratified for sex. The number of male and female PACG patients and controls for each study is clearly shown, together with the number of *UBOX5* carriers in each study strata.

5) The classifications of pathogenic variants need justification. The authors should use more rigorous tools like the REVEL scores rather than Polyphen2. Since REVEL scores encompasses 13 different tools (including Polyphen2), it is more reliable in terms of predictions for missense variants. I would suggest using REVEL scores for reclassifying the observed variants.

Response: We take onboard the Reviewer's suggestion and attempted to use REVEL scoring (whose scores encompasses the following 13 functional effect prediction algorithms: (MutPred, FATHMM, VEST, PolyPhen, SIFT, PROVEAN, MutationAssessor, MutationTaster, LRT, GERP, SiPhy, phyloP, and phastCons) for *UBOX5* in the discovery stage (4,667 PACG patients and 5,473 unaffected controls) to evaluate its performance.

Using the suggested REVEL score threshold of >0.5 (variants with scores above the threshold will be considered as pathogenic, Ioannidis NM et al., *Am J Hum Genet* 2016; 99:877–885), only 2 variants (Y493S, present in 9 PACG cases and 1 control, and R301Q, present in 1 PACG cases and in 0 controls) in the discovery exome sequencing study fulfilled such stringent criteria. In the validation study, variant P141R had REVEL score >0.5. This variant was observed in the UKBiobank study, and was seen in 1 out of 1,759 PACG cases (0.057%) and in 12 out of 467,880 controls (0.0026%), suggesting a more than 20-fold enrichment in PACG cases compared to controls. Although all three variants (Y493S, R301Q, and P141R) were found to be functionally deficient via biological assays, there remained many other functionally

deficient variants with benign REVEL scores (i.e. <0.5, see subsequent **Figure**). Thus, the analysis showed that REVEL appeared to be too strict in identifying qualifying variants, resulting in severely limited statistical power.

Gel quantitation data measuring the functional activity of *UBOX5* protein-altering variants to ubiquitinate BIP. Data are presented in triplicates. Intensity of the HA band was quantified and then normalized to the MYC signal of the same elute sample in the same gel. MYC signal was obtained after stripping the initial HA immunoblot and re-probing with MYC antibody. Wild-type *UBOX5* (green bar) was measured independently 15 times to provide a confidence estimate of wild-type *UBOX5* activity, so that functionally deficient *UBOX5* variants (defined as <80% activity of wild-type *UBOX5*) can be more confidently identified. Variants in blue bars represent normally functioning *UBOX5* alleles. A total of 42 *UBOX5* variants were tested, comprising 35 variants detected in the initial study, and 7 variants in the validation study from Italy and Pakistan. Representative Western Blot gel photographs are presented as Supplementary Figure 7. The REVEL score for each variant resulting in amino acid substitutions are appended. No scores are provided for the 2 synonymous variants (S) and one deletion (D) variant. Variants with REVEL score >0.5 are in red.

We next evaluated the correlation between REVEL scores (the higher the score, the more likely a variant is predicted to impair protein function), with REVEL score >0.5 being indicative of a genetic variant being 'damaging' (Ioannidis NM et al., *Am J Hum Genet.* 2016; 99(4):877–88). Although there were 5 variants with REVEL scores >0.4 there were functionally impaired, the functional status of the rest could not be reliably classified by the REVEL score (see subsequent **Figure**).

Correlation between UBOX5 variant REVEL score (horizontal axis) and UBOX5 variant functional activity (vertical axis). Whereas variants with REVEL scores of >0.4 / >0.5 were functionally impaired relative to wild-type UBOX5 protein, the REVEL scores of all other variants (those with REVEL scores <0.4) were similar between functionally normal and functionally impaired variants.

The data showed that a balance needs to be struck between the ultra-strict REVEL composite score from multiple bioinformatic algorithms (high specificity, but low sensitivity; struggles to achieve statistical significance despite a high odds ratio of 5.71) and functional biological testing (with higher sensitivity but lower specificity; performs best in statistical tests).

6) While the data was consistent with regards to the UBOX5 qualifying variants across all the discovery populations, were there variant(s) of other associated gene(s) that were overrepresented amongst these cases?

Response: Previous GWAS studies on PACG revealed significant association at 8 loci. There were 38 genes underlying the 8 loci previously reported to show genome-wide significant association with PACG risk. None of these 38 genes showed even nominally significant differential rare variant burden between cases and controls (**Supplementary**

Table 6), suggesting little overlap between common and rare variant genetic architecture for this disease.

7) It would also be useful to know if the carriers of *UBOX5* qualifying variants also harbored the GWAS associated variants? Did the presence of an associated common variant along with the *UBOX5* rare variant increase the susceptibility to APAC and/or PACG? Likewise, did the presence (or absence) of additional variants of other associated genes along with *UBOX5* variants exhibit enhanced risk of the disease? Clinically, did the presence of multiple alleles (rare and common) impact disease severity in terms of prognosis? How were the carriers of *UBOX5* qualifying variants different in terms of their disease manifestations compared to those who did not harbor these variants?

Response: Following on from point #6, the data is suggestive that common and rare genetic variant architecture for PACG risk are likely independent from one another. In addition, the very modest effect sizes of common variants (Odds Ratios of between 1.14 to 1.42; Khor CC et al., *Nat Genet* 2016; 48:556-62) are most unlikely to interact with carriage of rare *UBOX5* variants. Our analysis did not detect significant enrichment of *UBOX5* variants with analysed GWAS-associated SNPs, and overlapping numbers are too small to make meaningful comparisons in terms of disease manifestations.

8) Since the variants are rare and present in lesser numbers across populations, it is helpful to get an overall estimate of their burden by adding them (as was done by the authors). But biologically, the types of these variants would exhibit exclusive cellular and molecular functions based on their protein domains and specific interactions. This possibility needs to be discussed in detail.

Response: We agree with the Reviewer on the concept that different types of variants might exhibit different cellular and molecular functions depending on location of the rare variant with regards to protein domain and even interactions.

However, we know that *UBOX5* functions to ubiquitinate BIP. Thus, we directly analysed the impact of the different *UBOX5* variants on their activity to ubiquitinate BIP. In these experiments, we measured the ability of the different *UBOX5* variants to ubiquitinate BIP relative to wild type *UBOX5* and identified the functionally deficient variants. We did not dissect the possible, different molecular mechanisms from which functional deficiency might arise, as this is beyond the scope of the current study.

9) It is unlikely that variants in UBOX5 would be solely responsible for PACG. A complete profile of the downstream targets of UBOX5 should be provided through a network analysis. This would also demonstrate the potential functional interactions of other associated genes (both obtained from WES and those identified from earlier GWAS) along with UBOX5.

Response: We agree with the Reviewer that variants in UBOX5 are not solely responsible for PACG.

There were no known validated downstream targets of UBOX5 in the literature, and we were mindful that interaction between UBOX5 and another candidate protein does not imply a functional interaction, given that UBOX5 is an ubiquitin E3 ligase. This was the reason why we attempted to identify substrates of UBOX5 using an unbiased substrate trapping approach. The following proteins were observed to be maximally enriched (100x abundance ratio) and highest-scoring (top-10 Sequest HT score) across two biological replicates: HSPA1B, HSPA8, HSPA5 (BIP), IRS4, UBB, UBC, and MAGED1 (see **Supplementary Table 3**). Of these 7 proteins, HSPA1B, HSPA8, and HSPA5 (BIP) show markedly more binding to the UBOX5-UBD fusion protein (as defined by Score Sequest HT: The protein score which is calculated by summing the individual scores of each peptide. The higher this score, the higher the individual scores of the peptides, and thus the better the identification. SEQUEST HT is the name of the employed search engine) compared to IRS4, UBB, UBC, and MAGED1.

Because all three heat shock proteins are involved in endoplasmic reticulum (ER) stress and the unfolded protein response (UPR), this finding suggests that UBOX5 could be acting through the UPR pathway.

10) The functional basis of UBOX5 being directly involved in PACG is somewhat “over-stated” and much of the discussion is speculative. There are repeated statements inferring that a specific function for UBOX5 has been identified – which was not.

Response: Substrate trapping assays using UBOX5-UBD fusion bait coupled to Mass-Spectrometry, followed by subsequent demonstration of **a)** direct interaction of UBOX5 with BIP (**Figure 2A**) and ubiquitination of BIP in cells when UBOX5 was expressed (**Figure 2B**) suggest that BIP is a bona fide target of UBOX5. BIP is known as an important sensor of endoplasmic reticulum stress in the cell, and an initiator of the unfolded protein response pathway. New data in the revised manuscript suggests that by ubiquitinating BIP, UBOX5 increases the half-life of BIP.

We agree with the Reviewer and have modified the manuscript to adhere strictly to conclusions backed by evidence. **Specifically, the speculative parts of the discussion have been deleted in the revised main text, and we do not make any claims about UBOX5 being directly involved in PACG. Our statements are now softened to “suggest” the involvement of UBOX5 in PACG.**

I would have expected more mechanistic details from this consortium as additional downstream targets could have been analysed under a specific model system.

Response: In keeping with our response to point #9 to this Reviewer, the top 3 molecules observed to bind to UBOX5 from the substrate trapping assay via co-immunoprecipitation of targets with the UBOX5-UBD bait protein were HSPA8, HSPA1B, and HSPA5 (BIP)(see **Supplementary Table 3**). All three were heat shock proteins (HSPs) and all three proteins are involved in endoplasmic reticulum (ER) stress and the unfolded protein response (UPR). This preponderance of molecules involved in ER stress and UPR provides strong evidence suggesting that UBOX5 is associated with PACG via this specific pathway.

We selected BIP for functional assays due to its upstream sensor role in proteostasis (Kopp MC et al., *Nat Struct Mol Biol*; 2019; 26:1053-1062), as evidenced by its presence in the endoplasmic reticulum (Bonam SR et al., *Cells* 2019; 8(8):849; McNulty S et al., *Immunology* 2013 Aug;139(4):407-15). We observed in the initial manuscript submission that functionally deficient UBOX5 did not ubiquitinate BIP to the same extent as wild-type BIP (**Table 1, Figure 4 and Supplementary Figure 7**).

To further define the relationship between UBOX5 and BIP, as it remained unclear how UBOX5 could affect BIP function, we now include additional experiments showing that the presence of UBOX5 increases the half-life of BIP under conditions of ER stress compared to absence of UBOX5, as shown in the figure on the next page:

UBOX5 increases the half-life of BIP in ER stress conditions. Top: Pulse chase of BIP with or without expression of UBOX5 in HEK293 cells. HEK 293 cells were transfected with UBOX5 or empty vector. 24h later, cells were treated with a pulse of 0.25 μ M Thapsigargin for 2 hours, and cycloheximide (150 mM) was added. Cells were collected at indicated time points for immunoblotting. GAPDH was used as loading control. Positions of molecular Weight standards are indicated on the left.

Bottom: Quantitation of BIP band intensities normalized to GAPDH intensity for every time point. Intensities are shown as fold changes compared to normalized BiP intensity at t=0 of pulse chase.

Conversely, functionally deficient UBOX5 alleles did not increase the half-life of BiP, as shown in the following figure:

Functionally deficient UBOX5 variants do not increase BiP half-life compared to wild-type UBOX5. Pulse chase of BiP in the presence of wildtype UBOX5 or variants, in HEK293 cells. HEK 293 cells were transfected with wildtype UBOX or indicated variants. 24h later, cells were treated with a pulse of 0.25 μ M Thapsigargin for 2 hours, and cycloheximide (150 mM) was added. Cells were collected at indicated time points for immunoblotting. GAPDH was used as loading control. UBOX5 expression was verified as indicated. GFP, expressed from a separate locus in the vector used, was also used to verify success of transfection.

This is an important result because BiP was recently shown to be short-lived (Shim SM et al., *Sci Signal* 2018 Jan 2;11(511):eaan0630) and there is a possibility that ubiquitination of BiP might lead to increased turnover, in turn contradicting our hypothesis. Because BiP has been well-described to have cytoprotective properties in responding to ER stress (Kudo T et al., *Cell Death Differ.* 2008; 15(2):364-75; James AW et al., *Neurochem Int* 2023 Oct;169:105573; Morris JA et al., *J Biol Chem.* 1997; 272(7):4327-34; Wang J et al., *eLife.* 2014 Jul 22; 3:e03496), and has a key role in protein quality control within the cell (Chen X et al., *Signal Transduct Target Ther* 2023; 8(1):352), the observation that UBOX5 increases the half-life of BiP provides additional support that UBOX5 – BiP may have cytoprotective properties. BiP is induced during ER-stress as a means to increase protein folding capacity of the cell. Increased half-life under these conditions would enable cells to further boost their proteostatic capacity, enabling a more efficient resolution of ER stress and return to homeostasis.

We have now revised the manuscript by adding these sentences into the discussion section: “Firstly, the ubiquitination of BiP by UBOX5 appeared to increase the half-life of BiP. In agreement with prior reports (Shim SM et al., *Sci Signal* 2018 Jan 2;11(511):eaan0630) that BiP has a short half-life, we find that co-expression of UBOX5 led to decreased BiP protein turnover. BiP senses and responds to ER stress by inducing the unfolded protein response pathway. This pathway is conserved in mammalian cells (Fribley A et al *Methods Mol Biol.* 2009; 559:191–204) and in plants (Liu Y et al., *Front Plant Sci.* 2022 Oct 7;13:1019414), and cells with pathway

deficiencies were significantly less able to survive stressful stimuli, suggesting that activation of the unfolded protein response pathway in response to ER stress confers cytoprotective properties. Protein abundance of BIP is induced during ER stress to increase cellular protein folding capacity. Increasing its half-life would thus further increase cellular protein folding capability under these conditions, and lead to a rapid resolution of ER stress and return to homeostasis.”

Since PACG involves structural defects and IOP changes may occur independently of these defects, it would have been worthwhile to assay the interactions of UBOX5 and BIP and other targets in presence and absence of elevated IOP.

Response: Unfortunately, we do not have an IOP model (either cell-based or animal) available for assaying UBOX5 and BIP. However, it has been known that IOP increases endoplasmic reticulum stress in cells (Kroeger H., et al., *FEBS J* 2019; 286: 399-412; Doh, SH et al., *Brain Res* 2010; 1308:158-66), leading to the activation of the unfolded protein pathway.

We thus induced ER stress using thapsigargin (thapsigargin also activates the UPR pathway; Yoshino H et al., *Mol Med Rep* 2017;15:2867-2872) and assayed the ability of UBOX5 to ubiquitinate BIP in the presence and absence of thapsigargin. We observed that the presence or absence of thapsigargin does not affect the ability of UBOX5 to ubiquitinate BIP (**Figure 3C**, lanes 4 and 8, also shown in following figure for ease of reference), suggesting that the ability of UBOX5 to ubiquitinate BIP was not dependent on cellular stress. Native UBOX5 was however found to be induced by these ER stressors (Fig. 3A). Given that most protein translation is strongly inhibited under conditions of ER stress (Wang M and Kaufman RJ, *Nature* 2016; 529:326-35), the observed induction of UBOX5 expression strongly suggests that UBOX5 could be involved in modulating BIP under these conditions.

The ability of UBOX5 to ubiquitinate BIP was not dependent on cellular stress. (Upper panel) MYC-tagged BIP, empty vector, UBOX5, or HA-tagged ubiquitin was co-transfected into HEK293 cells in the indicated combinations. 24 hours later, cells were treated with 0.7 μ M Thapsigargin (TG) or DMSO for 16 hours. TG is a known inducer of endoplasmic reticulum stress. Cells were then harvested and a MYC immunoprecipitation was performed on the input lysates. Eluates were immunoblotted with antibodies against HA to assess the extent of BIP ubiquitination. The membrane was then stripped and a MYC immunoblot was performed to assess immunoprecipitation efficiency. The degree of ubiquitination of BIP did not appear to differ with the addition of TG (lanes 4 and lanes 8).

(Lower Panel) UBOX5 or its empty vector contains a GFP open reading frame, which allows for assessment of transfection efficiency by assessing GFP abundance in input lysates. GAPDH was used as loading control.

Of the 3 top proteins binding to UBOX5 emerging from the substrate trapping assay (**Supplementary Table 3**, HSPA1B, HSPA8, and HSPA5 (BIP)), BIP is known to be located in the endoplasmic reticulum (ER) and is most proximal to the source of ER stress. Thus, assaying BIP will likely yield highest signal-noise ratio compared to HSPA1B or HSPA8.

This would have also implied if structural abnormalities had complex (or no) relationships to IOP. Localizations of UBOX5 in the relevant ocular tissues is only suggestive of its involvement. But unless experiments are designed to demonstrate that perturbations of its normal functions and interactions lead to PACG or its the endophenotypes, and that these are reversed through appropriate rescue mechanisms, its causal role in PACG pathogenesis would remain elusive.

Response: We fully agree with this Reviewer. We humbly discuss here that perturbing UBOX5 using animal models of PACG followed by studying rescue mechanisms might be content for future work. We ask to be allowed to restrict data presentation in the current manuscript to the genetic study, substrate trapping with immunoprecipitation Mass-Spectrometry, establishment of the functional biological assay for UBOX5, and the biological testing of >30 independent alleles of UBOX5. All pieces of data converge to suggest that functionally deficient UBOX5 is associated with increased risk of PACG.

Some minor concerns pertain to few typos and syntax errors and use of “95%CI” and not “95%ci” throughout the text.

Response: We have now corrected these.

Reviewer #2 (Remarks to the Author)

The manuscript presents a well-designed and comprehensive study that investigates the role of UBOX5 variants in Primary Angle-Closure Glaucoma (PACG). By combining whole exome sequencing (WES), replication studies, and functional assays, the study provides novel insights into the genetic underpinnings of PACG and highlights UBOX5 as a gene of interest. The strengths of the study include its rigorous methodology, large sample sizes, and the functional validation of UBOX5 variants, which add substantial value to the genetic association findings. However, there are areas where the manuscript could be strengthened, particularly in terms of mechanistic insights and clarity on variant selection. Below are specific comments and suggestions for improvement.

Response: We are much grateful for the generous summary by this Reviewer and will work towards addressing all specific comments and suggestions.

Major Comments

1. The authors have discussed the limitations in the manuscript text, which is reflected in the Figure 1 where in Post-hoc analyses on sex stratification and risk of angle-closure glaucoma except the Japanese cohort all other cohorts are not showing significant risk of having the diseases especially in men from all the four sites.

Response: We agree with this Reviewer. The sex-stratified post-hoc analysis shown in **Figure 1** did not show significant association in men for primary angle-closure glaucoma (PACG)(Odds Ratio = 1.66, 95% confidence interval = 0.95-2.91) as well as for acute angle-closure glaucoma (Odds Ratio = 0.68, 95% confidence interval = 0.21-2.25).

This might be due to a smaller sample size in men, as PACG is more prevalent in women (Zhang N et al., *Front Med (Lausanne)* 2021 7:624179). In our study, out of the 4,667 persons with PACG that were analysed in the discovery exome sequencing study, only 1,773 were men (38 percent). Out of the 1442 persons with acute angle-closure glaucoma, only 418 were men (29 percent). Thus, the substantially lower number of men with PACG may suggest lower statistical power. We have discussed this potential issue in the revised manuscript text: "in keeping with published observations, we found PACG to be more prevalent in female participants compared to male participants. The odds ratio of UBOX5 carriers were observed to be higher in females compared to males. Out of the 4,667 persons with PACG that were analyzed in the discovery exome sequencing study, only 1,773 were men (38 percent), and out of the 1442 persons with acute angle-closure glaucoma, only 418 were men (29 percent). Thus, the substantially lower number of men with PACG may suggest lower statistical power to detect an effect, rather than it being a true sex-specific effect."

2. While the study identifies UBOX5 variants as significant contributors to PACG, the mechanistic link between UBOX5's ubiquitination of BIP and PACG development is not fully explored.

Response: We agree with the Reviewer. In this revised manuscript, we perform additional experiments to go one step further in terms of mechanistic links. We were able to show that the presence of UBOX5 significantly increases the half-life of BIP compared to absence of UBOX5 under ER stress conditions. We observed that at 2-hour, 4-hour, and 6-hour time-points, the presence of UBOX5 resulted in less turnover of BIP compared to the absence of UBOX5 (empty vector, see following figure).

UBOX5 increases the half-life of BIP. Pulse chase of BIP with or without expression of UBOX5 in HEK293 cells. HEK 293 cells were transfected with UBOX5 or empty vector. 24h later, cells were treated with a pulse of 0.25 μ M Thapsigargin for 2 hours, and cycloheximide (150 mM) was added. Cells were collected at indicated time points for immunoblotting. GAPDH was used as loading control. Positions of molecular Weight standards are indicated on the left. Bottom: Quantitation of BIP band intensities normalized to GAPDH intensity for every time point. Intensities are shown as fold changes compared to normalized BIP intensity at t=0 of pulse chase.

And whereas variant UBOX5 that was functionally normal (variant D33N, see **Figure 4**, main text) increased the half-life of BIP to a similar extent as wild-type UBOX5, functionally deficient UBOX5 variants (K291R, R301Q, and S465C) did not increase the half-life of BIP.

Functionally deficient UBOX5 variants do not increase BIP half-life. Top: Pulse chase analysis of BIP in the presence of wildtype UBOX5 or variant UBOX5 (D33N, K291R, E301Q, and S465C), in HEK293 cells. HEK 293 cells were transfected with wildtype UBOX or indicated variants. 24h later, cells were treated with a pulse of 0.25 μ M Thapsigargin for 2 hours, and cycloheximide (150 mM) was added at time=0. Cells were then subsequently collected at indicated time points (in hours) for immunoblotting. GAPDH was used as loading control. UBOX5 expression was verified as indicated. GFP, expressed from a separate locus in the vector used, was also used to verify success of transfection. Bottom: Quantitation of BIP band intensities in cells expressing indicated UBOX5 variants, normalized to GAPDH intensity for every time point. Intensities are shown as fold changes compared to normalized BIP intensity at t=0 of pulse chase.

BIP is an inducer of the unfolded protein-response (UPR), triggered when endoplasmic reticulum (ER) stress occurs (Fribley A et al., *Methods Mol Biol.* 2009; 559:191–204). A normally functioning UPR is important, as cells with deficiencies in the UPR pathway were significantly less able to respond to (and thus survive) stressful stimuli.

Our experiments show that both UBOX5 and BIP were enriched in the endoplasmic reticulum fraction of the cell (**Figure 3B**, main text, and shown in response to subsequent comment #7 by this Reviewer). The ER plays an important role in maintaining cellular homeostasis. Noxious external stimuli triggers ER stress and the unfolded protein-response, which can lead to cell death (Zhang J et al., *Cell Death Dis* 2022; 13:1051; Iurlaro R et al., *FEBS J* 2016; 283(14):2640-52). Thus, a normally functioning UPR has cytoprotective functions. Because UBOX5 and BIP are part of the UPR, it is possible that a normally functioning UBOX5 – BIP process could have protective roles.

Based on our data, the normal function of UBOX5's ubiquitination of BIP appears to increase its half-life, thus preserving its role in the ER stress – UPR pathway. Functionally deficient UBOX5 does not appear to increase the half-life of BIP and thus may result in higher vulnerability to cell stress and death. This is an important result because BIP was recently shown to be short-lived (Shim SM et al., *Sci Signal* 2018 Jan 2;11(511):eaan0630) and there is a possibility that ubiquitination of BIP might lead to increased turnover, in turn contradicting our hypothesis. Because BIP has been well-described to have cytoprotective properties in responding to ER stress (Kudo T et al., *Cell Death Differ.* 2008; 15(2):364-75; James AW et al., *Neurochem Int* 2023 Oct;169:105573; Morris JA et al., *J Biol Chem.* 1997; 272(7):4327-34; Wang J et al., *eLife.* 2014 Jul 22; 3:e03496), and has a key role in protein quality control within the cell (Chen X et al., *Signal Transduct Target Ther* 2023; 8(1):352), the observation that UBOX5 increases the half-life of BIP provides additional support that UBOX5 – BIP may have cytoprotective properties. BIP is induced during ER-stress as a means to increase protein folding capacity of the cell. Increased half-life under these conditions would enable cells to further boost their proteostatic capacity, enabling a more efficient resolution of ER stress and return to homeostasis.

3. The major criticism of the manuscript lies upon the choice of a generic cell line like HEK293 and overexpression of proteins using transient transfection. The nature of intricate cellular and biochemical analyses depicted in this manuscript requires finer experiments with stable cell lines and validation in an in-vivo system.

Response: The Reviewer raises a very helpful and valid point. Whereas we agree that the choice of a generic cell line such as HEK293 and overexpression of proteins using transient transfection is a major limitation, we humbly argue that this limitation is mitigated due to the biological properties of the BIP molecule, which is found to be expressed ubiquitously due to its role in the unfolded protein pathway. This role is

conserved in mammalian cells (Fribley A et al., *Methods Mol Biol.* 2009;559:191–204) and in plants (Liu Y et al., *Front Plant Sci* 2022; 13:1019414).

It has been reported that BIP protects the cell against a myriad of insults and along with other components of the UPR machinery are highly conserved across species. The UPR is an ancient cellular stress pathway, conserved from yeast to humans (Lewy TG et al., *Yale J Biol Med.* 2017 Jun 23;90(2):291–300). The high conservation of BIP across species is independently documented (Wang J et al., *Gene.* 2017; 618:14–23), and its fundamental role (Hetz C and Papa FR. *Mol Cell.* 2018; 69:169-181) renders it highly unlikely that its activity and properties will change in different cell lines. In this context, we argue that the results from the HEK293 cell line will reflect results from other cell types as well.

Considering the evidence presented (as a response to point #2 raised by this Reviewer), together with additional evidence showing that UBOX5 is indeed expressed in the iris and retinal ganglion cells (shown in **Supplementary Figures 2, 3, 4, and 5**), two sites important for PACG, **we ask to be allowed to defer further mechanistic studies to future work.** It is not possible to elucidate the exact mechanism linking UBOX5 deficiency and PACG risk in cell lines, as it will necessitate an animal model. The creation of such an animal model is likely beyond scope of this manuscript.

4. In Figure 2A, FLAG is used for IP but no IB: FLAG is shown. IB:HA should also be included here to shed more light on ubiquitinated BiP.

Response: We agree with the Reviewer, and now show blots with IB:FLAG and IB:HA as suggested as **Supplementary Figure 12** in the revision. We append it in the following figure for ease of reference:

FLAG-tagged UBOX5 (3 lanes on the right) or FLAG-tagged UBOX5-UBD construct (4 lanes on the left) and HA-tagged ubiquitin was co-transfected into HEK293 cells. After the first immunoprecipitation of lysates by anti-FLAG antibody, 20% of eluates was kept for analysis. Eluate from the UBOX5-UBD was further immunoprecipitated with anti-HA antibody to enrich for ubiquitinated proteins. To serve as antibody specificity control (negative controls), lysates were mock immunoprecipitated with mouse immunoglobulin (IgG). Immunoblotting of HA or FLAG was performed on inputs and eluates as indicated. Bands corresponding to ubiquitinated UBOX5 and ubiquitinated UBOX5-UBD chimeric protein is indicated by vertical lines, while the unmodified proteins are indicated by arrows on the right. Positions of the molecular weight markers are indicated by arrows on the left. IB:HA showed smeared bands because a multitude of ubiquitinated proteins (and not just BiP) were bound to it.

In the bottom panel lane 2, it is not clear what are lower bands in the 1st IP input lane.

Response: The lower bands in the first IP input lane were bands corresponding UBOX5 – UBD fusion protein bands. These could correspond to degradation species that were visualised because cells were treated with MG132, a proteasome inhibitor before harvesting (do see the following figure for ease of reference).

5. In Figure 2B, are myc-tagged BIP, UBOX5 and HA-tagged ubiquitin in same plasmid backbone? If yes, is it the only empty vector that has been taken as control? If not, where are other empty vector controls for this experiment? Where are the controls for Thapsigargin and MG132? Those results should also be included here.

Response: In Figure 2B, MYC-tagged BIP, UBOX5 and HA-tagged ubiquitin were in 3 separate plasmids. To clarify, all three were not in a single plasmid backbone.

Because domain analysis of UBOX5 suggested that it has E3-ubiquitin ligase activity, we asked whether UBOX5 could ubiquitinate BIP. Lanes 9 and 10 of Figure 2B serves as a negative control for the entire experiment, because in the absence of HA-tagged ubiquitin, no ubiquitinated BIP could be detected on the blot (do refer to the following figure for ease of reference). In place of HA-tagged ubiquitin, we used the empty vector (pCI neo) without the HA tagged ubiquitin.

Lanes 1-4 serves as a negative control for UBOX5, for it showed that in the absence of UBOX5, despite the presence of BIP and ubiquitin, no ubiquitinated BIP can be detected on the blot. In place of the UBOX5 expressing plasmid, we used the modified empty vector pcDNA 3.1 with a GFP open reading frame replacing the neomycin marker as a marker for transfection (**Figure 2C**, main text). This contrasts with the presence of UBOX5 in lanes 6 and 8, whereby ubiquitinated BIP was clearly blotted with IP:MYC (for BIP) and IP:HA (for ubiquitin).

The presence (lane 8) or absence (lane 6) of MG132 (a proteasome inhibitor) did not affect the ability of UBOX5 to ubiquitinate BIP. DMSO was used as carrier control in the MG132 negative lanes. MYC-tagged BIP was used as bait for the experiment, no negative control is needed because we are trying to determine whether UBOX5 ubiquitinates BIP. Without BIP, there will not be any result. However, we probed for MYC on the same blot after probing for HA to control for immunoprecipitation efficiency, which should be comparable between immunoprecipitation samples. We thereafter used the MYC signal as a normalizing factor to compare and quantify the HA signal intensity of the monoubiquitinated BIP species between different UBOX5 variants tested (now shown in **Supplementary Figure 7** in the form of Western blots and **Figure 4** after gel quantitation was performed, in the revised manuscript).

Also How did the authors tackle of co-transfecting three different constructs transiently and making sure that the cells are receiving all three constructs. Besides, there could be payload issue too.

Response: Ubiquitination of BIP will only occur when HA, BIP and UBOX5 are present in the same cell, so the generated data arises from successful ubiquitination events that could have only occurred when all three plasmids are transfected into the same cell. To demonstrate transfection efficiency, the plasmid encoding UBOX5 contains a GFP locus driven by SV40 promoter, and transfection efficiency was directly assayed by probing for both GFP and UBOX5 (**Figure 2C**). Before analysis, we will ensure that the GFP immunoblot signal between UBOX5 variants is always uniform, to ensure transfection efficiency is comparable between different samples (**Supplementary Figure 7**) expressing different UBOX5 variants. BIP is MYC-tagged, and is directly immunoprecipitated from input lysates. Thus a MYC immunoblot is always generated after stripping the HA immunoblot, and always accompanies elute and input lanes to ensure that MYC signal intensities in the elute lanes to be similar to ensure the same immunoprecipitation efficiency between different immunoprecipitation samples. In our attempts to quantify the intensity of monoubiquitinated BIP, we always normalize the intensity of the monoubiquitinated BIP to the MYC intensity to account for differences in immunoprecipitation efficiency.

Lastly, the transfection efficiency of ubiquitin-tagged HA is unlikely to change between variant UBOX5 and wildtype UBOX5, since the variants are single-nucleotide substitutions and have the same amino acid chain length as wild-type reference, and the same amounts of HA-tag ubiquitin are transfected. This is another advantage of using limited point mutation scanning to dissect protein function, as deletion constructs would have differing transfection efficiencies due to differing cDNA sizes (see response to Point #9)

6. In Figure 3A, gel quantitation data is required for the immunoblot.

Response: We agree with the Reviewer and now present gel quantitation data with necessary triplicates and error bars for the **revised Figure 3A**. Do find it appended here:

UBOX5 and BIP are both induced by ER stress. NIH3T3 cells were treated with the indicated ER stress inducers tunicamycin (Tu) and thapsigargin (Tg). (Top) Endogenous UBOX5 mRNA abundance was quantified by qPCR, normalised against mouse beta-actin transcript. Relative fold change of transcript is reported against control DMSO treatment. (Middle) Protein abundance of endogenous mouse ubox5 in ER-stressed NIH3T3 cells. Beta-actin was used as loading control. The position of the 60 kDa molecular weight band is indicated by the arrow on the left. Of note, UBOX5 mRNA and protein is induced in response to ER stress. (Bottom): Densitometric quantitation of UBOX5 bands. UBOX5 band intensity of each sample is normalised to corresponding beta-actin intensity. The amount of TG or TU used is indicated on the x-axis. *: p<0.05; **: p<0.01; ***: p,0.001, Welch's t-test.

We have also performed gel quantitation with triplicates to other important blots as well, such as for **Figure 4**, as well as for the new **Figure 5** (pulse chase data on BIP half-life) in the revised manuscript.

The new **Figure 4** reflects gel quantitation data from immunoblots presented in **Supplementary Figure 7**. Here, the intensity of the monoubiquitination band was normalized to intensity of the MYC band in the same blot, to correct for differences in immunoprecipitation efficiency; GFP intensity, taken to be transfection efficiency was also performed on the input lysates and verified to be comparable to the different inputs which are transfected with different variants, before quantitation was commenced.

The activity of wild-type UBOX5 was measured independently 15 times to provide a confidence estimate of wild-type UBOX5 activity, so that functionally deficient UBOX5 variants can be more confidently identified.

This data is now shown as **Figure 4** in the revised manuscript main text (also appended here for ease of reference).

Gel quantitation data measuring the functional activity of UBOX5 protein-altering variants to ubiquitinate BIP. Data are presented in triplicates. Wild-type UBOX5 (green bar) was measured independently 15 times to provide a confidence estimate of wild-type UBOX5 activity, so that functionally deficient UBOX5 variants (defined as <80% activity of wild-type UBOX5) can be more confidently identified. Variants in blue bars represent normally functioning UBOX5 alleles, and variants in orange bars represent functionally deficient UBOX5.

7. In Figure 3B, the authors should show the status of endogenous UBOX5 in HEK293 cells.

Response: Endogenous UBOX5 (see **Figure 2C** in the main text, reproduced below for ease of reference) was undetectable in un-transfected HEK293 cells. Hence, the assay readouts were clean with no interference from endogenous UBOX5.

In addition, the authors should also include the nuclear fraction data immunoblotted with UBOX5, BIP, GAPDH and calnexin to establish the specificity of their fractionation method especially in the context of ER and nuclear fraction. They might choose to use LaminA/C as a marker for the nuclear fraction.

Response: We agree with the Reviewer and have included nuclear fractionation data immunoblotted with UBOX5, BIP, GAPDH and calnexin (shown in **Figure 3B** of the revised manuscript and appended in the following page for ease of reference). We observed Calnexin staining in the ER fraction as well as in the nuclear fraction (as Calnexin is known to associate with the outer nuclear membrane; Dreger M et al., *Proc Natl Acad Sci U S A.* 2001 Oct 9;98(21):11943-8).

Cellular Localization of UBOX5: Immunoblots of HEK293 cells transiently transfected with UBOX5 expression plasmid and treated with 0.7 μ M Thapsigargin for 16h. Cellular fractions are indicated above. Whole cell lysates were fractionated into cytoplasmic, endoplasmic reticulum (ER) fractions and nuclear fractions by stepwise centrifugation. Indicated antibodies are shown. Positions of molecular weight markers are indicated on the left with arrows. GFP is used as transfection control, Calnexin is used as fractionation control for ER and nuclear fraction; Histone H2B is used as fractionation control for nuclear fraction. GAPDH is used as fractionation control for cytoplasmic and nuclear fractions.

8. In Figure 3C, where is the empty vector control in the immunoblot analyses?

Response: Similar in response to Comment #5 by this Reviewer, MYC-tagged BIP, UBOX5 and HA-tagged ubiquitin are in 3 separate plasmids. Because UBOX5 has E3-ubiquitin ligase activity, we asked whether UBOX5 could ubiquitinate (deposit ubiquitin molecules on) BIP.

Lanes 9 and 10 serve as negative controls, where the absence of HA-tagged ubiquitin (with the corresponding empty vector pCI Neo used in place of the HA-tagged ubiquitin) results in no ubiquitinated BIP being detected on the blot (see appended Figure in the following page). Conversely, the absence of UBOX5 also resulted in no ubiquitinated BIP being detected on the blot (see lanes 2 and 6 for the eluate). The presence of UBOX5 in lanes 4 and 8 clearly resulted in ubiquitinated BIP being detected on the blot.

The presence or absence of Thapsigargin (TG), a trigger of endoplasmic reticulum stress, did not affect the ability of UBOX5 to ubiquitinate BIP (lane 4, without TG; and lane 8, with TG). DMSO was used as carrier control for thapsigargin.

9. The experiment depicted in Figure 4, does not necessarily prove UBOX5 is solely responsible for the physical interaction between BIP and ubiquitin. The authors should first perform an experiment by in-vitro pull-down to detect the domain in UBOX5 is responsible for binding/deposit of ubiquitin on BIP. Afterwards another experiment is required upon making deletion constructs of UBOX5 to prove that really perturbs the binding/deposit of ubiquitin on BIP.

Response: The role of UBOX5 in the physical interaction between BIP and ubiquitin was addressed in two independent experiments presented in the manuscript main text (**Figure 2B** and **3C**). In **Figure 2B**, the presence of ubiquitin and BIP but absence of UBOX5 (lanes 2 and 4) did not result in detection of ubiquitinated BIP via immunoblot. However, the addition of UBOX5 resulted in ubiquitinated BIP being clearly detected on immunoblot (lanes 6 and 8).

Similarly, in **Figure 3C**, the absence of UBOX5 (lanes 2 and 6) did not result in detection of ubiquitinated BIP via immunoblot. However, the presence of UBOX5 resulted in ubiquitinated BIP being clearly detected on immunoblot (lanes 4 and 8). The consistency of the result suggests that UBOX5 is a major requirement for BIP to be ubiquitinated.

Scrutinizing the UBOX5 literature in more detail, we note that in-vitro ubiquitination experiments have been previously performed (Hatakeyama S et al., *J Biol Chem* 2001; 276(35):33111-20), and it was shown that the UBOX domain was important for UBOX5 ubiquitination activity. Because it was well documented that E3 ligases tend to be promiscuous binders to E2s and their substrates in vitro (Yalçin Z et al., *Mol Cell Proteomics* 2023; 22:100548 and Iconomou M et al., *Biochem J* 2016; 473:4083-4101), we decided to find UBOX5 substrates in cells by using a previously

published substrate trapping approach (Watanabe M et al., *Commun Biol* 2020 Oct 20;3(1):592).

As identifying the correct E2 partner in cells that mediates ubiquitination of BIP was a non-trivial task beyond the scope of this paper, we decided instead to extend our initial observation that UBOX5 binds BIP in cells (**Figure. 2A**, main text) by performing immunoprecipitation of UBOX5 deletion mutants to determine which domain is important for UBOX5-BIP interaction, before subjecting them to the ubiquitin ligation assay (**Figure 4** and **Supplementary Figure 7** in the revised manuscript). We experienced extreme difficulty expressing these deletion mutants in cells. We initially made an entire series of deletion mutants with deletions of the UBOX domain, RING domain, as well as serial deletions of stretches of amino acids (e.g amino acids 1 to 100, 1 to 259, 100 to 259, and 100 to 422) (do see **panel A in the Figure on the following page**).

For all deletion constructs, despite verifying the success of transfection based upon GFP expression, we were unable to express these mutants to the level comparable to wild-type UBOX5. To rule out the loss of antibody epitope due to deletion of large regions of UBOX5 protein, we tagged the C terminus of UBOX5 with FLAG epitope, repeated the transfection experiments, and probed for expression using anti-FLAG antibody instead. Unfortunately, we were still unable to observe appreciable expression of UBOX5 deletion variant protein (**panel B in the Figure on the following page**). As these UBOX5 deletion variants could not be expressed in cells, we were unable to proceed further.

Nonetheless, naturally occurring point mutations that do not drastically alter protein structure (compared to the effect of the deletion variants described in the text above) are suitable substitutes to study E3 ubiquitin ligase activity of UBOX5 on BIP. The UBOX domain of the UBOX5 protein is located between amino acid positions 260 to 312. We observed 3 naturally occurring variants (K291R, R301Q and A311D) within this domain. All three variants showed significant functional impairment compared to wild-type UBOX5 (**Figure 4**, main text) and all three were found only in PACG cases (2 carriers for K291R, 1 carrier for R301Q, and 3 carriers for A311D) and not in controls (**Table 1**, main text). When considered together with prior evidence that the UBOX domain was essential for the E3 ubiquitin ligase activity of mammalian UBOX proteins (Hatakeyama S et al., *J Biol Chem* 2001; 276(35):33111-20), this further solidifies the notion that the association between functional impairment of UBOX5 and increased risk of PACG might be due to loss of UBOX5's E3 ubiquitin ligase ability to ubiquitinate BIP.

**Expression of UBOX5 deletion mutants in HEK293 cells. (A)** Top: The schematic of the UBOX5 protein is shown. The conserved UBOX and RING domains are indicated together with the amino acids positions which spans these domains. Bottom: UBOX5 deletion constructs together with wild-type UBOX5 and empty vector were transfected into HEK293 cells and immunoblots of harvested cell lysates were performed to verify expression of the variants in HEK293 cells. The N terminus variant deletions are indicated by their amino acid positions. GFP expression, driven by a separate locus in the same plasmid was used as a transfection control. GAPDH was used as a loading control. **(B)** C terminal FLAG tagged UBOX5 deletion constructs, wild-type UBOX5 as indicated, were transfected into HEK293 cells and an immunoblot was performed as (A).

Also, gel quantitation data with number of biological replicates with proper statistical analyses should also be included.

Response: We have now provided gel quantitation data with necessary triplicates and error bars. The activity of wild-type UBOX5 was measured independently 15 times to provide a confidence estimate of wild-type UBOX5 activity, so that functionally deficient UBOX5 variants can be more confidently identified. This data is now shown as **Figure 4** in the revised manuscript main text (and appended below for ease of reference). We have moved the Western Blot gel photographs to **Supplementary Figure 7** in the revision.

Gel quantitation data measuring the functional activity of UBOX5 protein-altering variants to ubiquitinate BIP. Data are presented in triplicates. Wild-type UBOX5 (green bar) was measured independently 15 times to provide a confidence estimate of wild-type UBOX5 activity, so that functionally deficient UBOX5 variants (defined as <80% activity of wild-type UBOX5) can be more confidently identified. Variants in blue bars represent normally functioning UBOX5 alleles, and variants in orange bars represent functionally impaired UBOX5 alleles. A total of 42 UBOX5 variants were tested, comprising 35 variants detected in the initial study, and 7 variants in the validation study from Italy and Pakistan.

10. In Table 1, since most of these mutations are rare in populations, CADD score is not enough. Ideally these should be tested using various bioinformatics tools such as SIFT, LRT, MutationTaster, Provean etc. and it should be considered deleterious when at least three of these tools designate the mutation as deleterious. The authors could consider including a matrix showing how their functional annotation is corroborating with the bioinformatics prediction tools and provide a graph where functionally deficient mutations will be assessed further with their prediction scores generated using different tools.

Response: We agree with this Reviewer, which was consistent with comment #5 from Reviewer 1. We now include predictions from the following 13 software into an integrated REVEL score (MutPred, FATHMM, VEST, PolyPhen, SIFT, PROVEAN, MutationAssessor, MutationTaster, LRT, GERP, SiPhy, phyloP, and phastCons; Ioannidis NM et al., *Am J Hum Genet.* 2016; 99(4):877–88).

Using the suggested REVEL threshold of >0.5 (Ioannidis NM et al., *Am J Hum Genet* 2016; 99:877–885), only 2 variants (Y493S, present in 9 PACG cases and 1 control, and R301Q, present in 1 PACG cases and in 0 controls) in the discovery exome sequencing study fulfilled such stringent criteria. Including just these 2 variants into the gene-based burden test for *UBOX5* yields an Odds Ratio of 5.7 (95% confidence interval of 0.73 – 44.7) and a *P*-value of 0.06. This manner of analysis would have resulted in *UBOX5* being missed as a susceptibility gene for PACG in the context of an exome-wide discovery screen.

Amongst the 42 *UBOX5* variants subjected to functional biological analysis, 3 variants had REVEL scores of >0.5 . These were Y493S, R301Q, (both discussed in the preceding paragraph) and P141R. *UBOX5* P141R was observed in the UKBiobank study, and was seen in 1 out of 1,759 PACG cases (0.057%) and in 12 out of 467,880 controls (0.0026%)(Odds Ratio = 22.2, 95% ci: 2.88 – 171). Although all three variants (Y493S, R301Q, and P141R) were also found to be functionally deficient, there remained many other functionally deficient variants with benign REVEL scores (i.e. <0.5 , see subsequent **Figure**). Thus, the analysis showed that REVEL appeared to be too strict in identifying qualifying variants, resulting in severely limited statistical power.

Gel quantitation data measuring the functional activity of **UBOX5** protein-altering variants to ubiquitinate BIP. Wild-type UBOX5 (green bar) was measured independently 15 times to provide a confidence estimate of wild-type UBOX5 activity, so that functionally deficient UBOX5 variants (defined as <80% activity of wild-type UBOX5) can be more confidently identified. Variants in blue bars represent normally functioning UBOX5 alleles, and variants in orange bars represent functionally impaired UBOX5 alleles. The REVEL score for each variant resulting in amino acid substitutions are appended. No scores are provided for the 2 synonymous variants (S) and one deletion (D) variant. Variants with REVEL score >0.5 are in red.

We next evaluated the correlation between REVEL scores (the higher the score, the more likely a variant is predicted to impair protein function, with REVEL score >0.5 being indicative of a genetic variant being 'damaging' (Ioannidis NM et al., *Am J Hum Genet.* 2016; 99(4):877–88). Although we observed at all 3 variants with REVEL score >0.5 had impaired protein function, there were many functionally impaired variants with low REVEL scores that were similar to functionally normal variants (see following Figure).

Correlation between UBOX5 variant REVEL score (horizontal axis) and UBOX5 variant functional activity (vertical axis). Whereas variants with REVEL scores of >0.4 / >0.5 are functionally impaired relative to wild-type UBOX5 protein, the REVEL scores of all other variants (those with REVEL scores <0.4) were similar between functionally normal and functionally impaired variants.

11. Further, the manuscript would benefit from additional experiments or a more detailed discussion on how this interaction could lead to the pathophysiological changes observed in PACG. Specifically, how does UBOX5-mediated mono-ubiquitination of BIP influence intraocular pressure, optic nerve damage, or other features of PACG? More clarity on this could significantly enhance the impact of the findings.

Response: Considering our evidence that functionally deficient UBOX5 variants were strongly associated with increased risk of PACG, we hypothesized that the normal function of UBOX5 might have cytoprotective properties (and thus protect the optic nerve from damage due to cellular stress).

Our follow up experiments described in the revised manuscript now show that the presence of wild-type UBOX5 increases the half-life of BIP compared to absence of

UBOX5. In contrast, loss-of-function UBOX5 variants do not have this BIP half-life increasing property. Allow us to consider 3 key additional pieces of information:

- A) Both UBOX5 and BIP are present in the endoplasmic reticulum (ER) fraction of the cell (**Figure 3B**),
- B) The role of the ER is to synthesize proteins (Schwarz DS and Blower MD. *Cell Mol Life Sci.* 2015; 73(1):79–94) necessary for cellular function
- C) Noxious external stimuli triggers ER stress and the unfolded protein-response (Zhang J et al., *Cell Death Dis* 2022 Dec 19;13(12):1051; Iurlaro R et al., *FEBS J.* 2016 Jul;283(14):2640-52).

Increased intraocular pressure exerts a noxious external effect on retinal ganglion cells, leading to glaucomatous optic neuropathy (Kroeger H et al., *FEBS J* 2019; 286(2):399-412. Doh SH et al., *Brain Res* 2010 Jan 13;1308:158-66. Hurley DJ et al., *Antioxidants (Basel)* 2022 Apr 29;11(5):886). As BIP is part of the unfolded protein response pathway activated in response to endoplasmic reticulum stress, a normally functioning UBOX5 – BIP signaling process could have a cytoprotective role.

12. The manuscript highlights certain UBOX5 variants for functional analysis but does not provide a clear rationale for their selection. It would be helpful to include more information on the criteria used to prioritize these variants over others. Were these variants chosen based on their frequency, predicted pathogenicity, or previous associations with PACG? Providing this context would improve the transparency of the study design and help readers understand the significance of the chosen variants.

Response: We selected 35 variants for initial functional testing. These variants were drawn from 5 participating countries (Singapore, Hong Kong, Japan, Vietnam, and the UK Biobank) with the largest number of PACG cases in the study. Due to their larger sample sizes, we posit that rare variant burden from these studies were more likely to be representative and less unlikely to be due to winner's curse. The following classes of UBOX5 variants were evaluated for their biological function:

a) variants observed to be enriched in PACG cases compared to controls, for example:

- p.Val109Ala, seen in 0.057% of cases but only in 0.00021% of controls.
- p. Pro110Leu, seen in 0.11% of cases but 0.0096% of controls
- p. Lys291Arg, found in 0.087% of cases but not in controls.

b) variants observed to be enriched in controls compared to PACG cases, for example:

- p. Lys18Arg, seen in 0.12% of cases, but in as many as 0.20% of controls.
- P. Ile206Lys, not seen in cases, but seen in 0.02% of controls.

c) variants seen equally in PACG cases and controls, for example:

- p. Asp33Asn. Seen in 0.038% of cases and 0.032% of controls.

- P. Pro498Pro, a synonymous variant, seen in 0.34% of cases and 0.37% of controls.

Laboratory tests suggest 20 of the 35 variants selected for initial biological testing to be functionally impaired (defined as <80% of wild-type UBOX5 activity; **Table 1, Figure 4 and Supplementary Figure 7**). Nineteen of the 20 (95%) functionally impaired variants appeared to be carried more often by persons with PACG compared to by unaffected individuals. Conversely, 10 of the 15 (66.7%) variants with normal function were carried more often by unaffected individuals compared to affected individuals (**Table 1**). This observation supports the notion that functionally impaired UBOX5 variants are significantly enriched in PACG patients compared to unaffected controls. The manuscript has been revised to make this explicit.

13. Although the authors address population stratification, the manuscript could benefit from a more detailed discussion of how genetic or environmental differences between the studied populations might influence the findings. For example, are there specific genetic factors in the Asian cohorts that could explain the stronger associations observed in these populations?

Response: It remains very challenging to propose how genetic or environmental differences between the studied populations might influence the findings due to the complex, multifactorial nature of PACG (He M et al., *Eye (Lond)* 2006; 20(1):3-12). He et al., described the following differences between Asians and Europeans:

- a) Presence of symptomatic disease
- b) Differences in iris anatomy
- c) Higher prevalence of myopia in Asians

without drawing firm conclusions.

Based on data from a very recent genome-wide association study meta-analysis of 6,034 PACG cases and 15,071 controls of Asian ancestry, together with 3,183 PACG cases, 773,214 controls of European ancestry (Luben R et al., Under invited revision at *Nature Communications*), there appeared to be very clear differences in common variant genetic architecture between Europeans and Asians (for example, *PLEKHA7*, *COL11A1*, *EPDR1*, being genome-wide significant loci for Asians, whereas *HERC2* and *LAMA2* were genome-wide significant loci for Europeans. Shared genome-wide significant loci between Asians and Europeans are *PXDNL* and *GLIS3*).

For rare *UBOX5* variants, the current study size does not provide sufficient statistical power to dissect the contribution of ancestry-specific *UBOX5* alleles to PACG risk, and we have revised the manuscript to make this explicit. We were unable to uncover specific genetic factors in the Asian cohorts that could explain the potentially stronger associations observed in these populations.

Additionally, discussing potential limitations related to population differences in the replication cohorts would add depth to the analysis.

Response: We agree with the Reviewer. We have inserted the following into the discussion section of the revised manuscript text: “Third, because the discovery analysis was performed in participants of Asian ancestry, but validation pursued in diverse panels of participants from around, pinning down the series of UBOX5 alleles that are causally related to PACG risk will require additional follow up.

Minor Comments

14. The statistical genomics methods used are robust, and the use of multiple cohorts enhances the study’s power. However, the manuscript would benefit from a clearer explanation of how the p-value threshold for significance was determined (e.g., why $P < 2.5 \times 10^{-6}$ was chosen for exome-wide significance). This would help readers better understand the stringency of the analysis.

Response: We agree with the Reviewer. The *P*-value threshold for significance was determined by adjusting the usual $P = 0.05$ threshold for statistical significance for 20,000 genes tested using the gene-based burden test. 0.05 divided by 20,000 genes gives $P < 2.5 \times 10^{-6}$ (Li Z et al., *JAMA* 2021; 325:753-764).

15. The identification of UBOX5 and its impact on BIP ubiquitination presents potential therapeutic targets for PACG. How do the authors envision translating these findings into targeted therapies?

Response: the UBOX5 – BIP axis could have cytoprotective / neuroprotective properties, as our experiments suggest that both UBOX5 and BIP were enriched in the endoplasmic reticulum fraction of the cell. Furthermore, the presence of wild-type UBOX5 increases the half-life of BIP, whereas functionally deficient UBOX5 does not.

The endoplasmic reticulum is known to an important role in maintaining cellular homeostasis. Noxious external stimuli triggers ER stress and the unfolded protein-response, which can lead to cell death (Zhang J et al., *Cell Death Dis.* 2022 Dec 19;13(12):1051; Iurlaro R et al., *FEBS J.* 2016; 283(14):2640-52).

In the context of intraocular pressure and glaucomatous optic neuropathy, a normally functioning UBOX5 – BIP process could have neuroprotective roles. Pharmacological approaches that augment the half-life of BIP, or further induce its expression and/or related approaches that increase cellular protein folding capability by inducing other protein chaperones might be a potential therapeutic approach (as we show in the revised manuscript that wild-type UBOX5 increases the half-life of BIP whereas functionally deficient UBOX5 variants do not increase the half-life of BIP).

16. The manuscript is generally well-written, but there are a few areas where clarity could be improved. For example, the description of the immunoprecipitation assays in the methods section could be simplified to make it more accessible to readers who may not be specialists in this technique. Additionally, some sentences in the discussion section are quite long and could be broken down for easier comprehension.

Response: We agree and have revised the description for the immunoprecipitation assay to explain the use of the FLAG tag, as follows: “The FLAG tag is a specific protein tag to which specific, high avidity monoclonal antibodies have been developed. This tag is particularly useful in assays that require specific recognition by antibodies, such as immunoprecipitation.”

Reviewer #3 (Remarks to the Author)

Association between PACG and rare, protein-altering genetic variants were found in UBOX5 through exome sequencing of thousands of patients and controls. Results were replicated in an additional study also of thousands of patients and controls. UBOX5 is expressed in the iris sphincter pupillae, optic nerve head, and throughout the retina. Co-IP and mass spec revealed that three heat shock proteins interacted with UBOX5. One of these, Binding immunoglobulin protein (BIP) was prioritized in particular due to its role as an inducer of the unfolded protein response pathway which are linked to glaucoma. Functional studies were done of 35 UBOX5 variants found to be enriched in PACG cases. 21/35 were found to be functionally impaired, and were carried more often in people with PACG versus controls. These functional studies were also replicated in additional cases. The authors suggest that the UBOX5-BIP pathway is involved in PACG pathogenesis.

Comments:

Authors are aware that the case control study design does not allow assignment of causal relationships. They are also aware of potential issues with determination of PACG status based upon diagnosis codes in the UK biobank rather than in a hospital. Studies were replicated at every stage, however, and this reviewer has confidence in the validity of both the genetic and biochemical analyses.

Response: Thank you for this generous assessment.

It is not really clear to me how these genes are linked to PACG, despite the authors assertion that UBOX5-BIP pathway is involved in glaucoma pathogenesis.

Response: The *UBOX5* gene was first linked to PACG via a significant statistical association ($P=1.25 \times 10^{-10}$) that was consistently observed across 14 independent sample collections. Carriers of rare, protein-altering genetic variants in *UBOX5* had 2.13-fold increased odds (95% confidence interval, 1.69 – 2.69) of PACG compared to non-carriers.

Using UBOX5-UBD as a bait to trap UBOX5 substrates, we performed an unbiased co-immunoprecipitation assay coupled with Mass-Spectrometry. The top 3 molecules found to bind to UBOX5 were heat shock proteins (HSPA1B, HSPA8, and HSPA5 [BIP]), and all three heat shock proteins are involved in endoplasmic reticulum (ER) stress and the unfolded protein response (UPR). The collective exome-sequencing and co-immunoprecipitation/substrate trapping data implicates the ER stress – UPR pathway as important in PACG pathogenesis.

Out of HSPA1B, HSPA8, and HSPA5 [BIP], we selected BIP for functional assays due to its upstream sensor role in proteostasis (Kopp MC et al., *Nat Struct Mol Biol*;

2019; 26:1053-1062), as evidenced by its presence in the endoplasmic reticulum (Bonam SR et al., *Cells* 2019; 8(8):849; McNulty S et al., *Immunology* 2013 Aug;139(4):407-15). We observed in the initial manuscript submission that functionally deficient UBOX5 did not ubiquitinate BIP to the same extent as wild-type BIP (**Table 1, Figure 4** and **Supplementary Figure 7**).

Consistent with our response to Reviewer #2, we perform additional experiments to go one step further in terms of mechanistic links. We were able to show that the presence of UBOX5 significantly increases the half-life of BIP compared to absence of UBOX5 under ER stress conditions. We observed that at 2-hour, 4-hour, and 6-hour time-points, the presence of UBOX5 resulted in less degradation of BIP compared to the absence of UBOX5 (empty vector, see following figure).

UBOX5 increases the half-life of BIP. Pulse chase of BIP with or without expression of UBOX5 in HEK293 cells. HEK 293 cells were transfected with UBOX5 or empty vector. 24h later, cells were treated with a pulse of 0.25 μ M Thapsigargin for 2 hours, and cycloheximide (150 mM) was added. Cells were collected at indicated time points for immunoblotting. GAPDH was used as loading control. Positions of molecular Weight standards are indicated on the left. Bottom: Quantitation of BIP band intensities normalized to GAPDH intensity for every time point. Intensities are shown as fold changes compared to normalized BIP intensity at t=0 of pulse chase.

And whereas variant UBOX5 that was functionally normal (e.g. D33N, see **Figure 4**, main text) increased the half-life of BIP to a similar extent as wild-type UBOX5, functionally deficient UBOX5 variants (K291R, R301Q, and S465C) did not increase the half-life of BIP.

Functionally deficient UBOX5 variants do not increase BIP half-life. Top: Pulse chase analysis of BIP in the presence of wildtype UBOX5 or variant UBOX5 (D33N, K291R, E301Q, and S465C), in HEK293 cells. HEK 293 cells were transfected with wildtype UBOX or indicated variants. 24h later, cells were treated with a pulse of 0.25 μ M Thapsigargin for 2 hours, and cycloheximide (150 mM) was added at time=0. Cells were then subsequently collected at indicated time points (in hours) for immunoblotting. GAPDH was used as loading control. UBOX5 expression was verified as indicated. GFP, expressed from a separate locus in the vector used, was also used to verify success of transfection. Bottom: Quantitation of BIP band intensities in cells expressing indicated UBOX5 variants, normalized to GAPDH intensity for every time point. Intensities are shown as fold changes compared to normalized BIP intensity at t=0 of pulse chase.

BIP is an inducer of the unfolded protein-response (UPR), triggered when endoplasmic reticulum (ER) stress occurs (Fribley A et al., *Methods Mol Biol.* 2009; 559:191–204). A normally functioning UPR is important, as cells with deficiencies in the UPR pathway were significantly less able to respond to (and thus survive) stressful stimuli.

Our experiments show that both UBOX5 and BIP were enriched in the endoplasmic reticulum fraction of the cell (**Figure 3B**, main text, and shown in response to subsequent comment #7 by this Reviewer). The ER plays an important role in maintaining cellular homeostasis. Noxious external stimuli triggers ER stress and the unfolded protein-response, which can lead to cell death (Zhang J et al., *Cell Death Dis* 2022; 13:1051; Iurlaro R et al., *FEBS J* 2016; 283(14):2640-52). Thus, a normally functioning UPR has cytoprotective functions. Because UBOX5 and BIP are part of the UPR, it is possible that a normally functioning UBOX5 – BIP process could have protective roles.

Based on our data, the normal function of UBOX5's ubiquitination of BIP appears to increase its half-life, thus preserving its role in the ER stress – UPR pathway. Functionally deficient UBOX5 does not appear to increase the half-life of BIP and thus may result in higher vulnerability to cell stress and death. This is an important result because BiP was recently shown to be short-lived (Shim SM et al., *Sci Signal* 2018 Jan 2;11(511):eaan0630) and there is a possibility that ubiquitination of BiP might lead to increased turnover, in turn contradicting our hypothesis. Because BIP has been well-described to have cytoprotective properties in responding to ER stress (Kudo T et al., *Cell Death Differ.* 2008; 15(2):364-75; James AW et al., *Neurochem Int* 2023 Oct;169:105573; Morris JA et al., *J Biol Chem.* 1997; 272(7):4327-34; Wang J et al., *eLife.* 2014 Jul 22; 3:e03496), and has a key role in protein quality control within the cell (Chen X et al., *Signal Transduct Target Ther* 2023; 8(1):352), the observation that UBOX5 increases the half-life of BIP provides additional support that UBOX5 – BIP may have cytoprotective properties. BiP is induced during ER-stress as a means to increase protein folding capacity of the cell. Increased half life under these conditions would enable cells to further boost their proteostatic capacity, enabling a more efficient resolution of ER stress and return to homeostasis.

The authors speculate that mono-ubiquitination of BIP by UBOX5 could aid in protecting retinal ganglion cells from ER stress and glaucomatous optic neuropathy, and functionally deficient UBOX5 might remove this layer of protection, but it is not clear to me how, or indeed if this occurs. Additional discussion and/or exploration of this connection would be useful.

Response: We now show experimental evidence that mono-ubiquitination of BIP by wild-type UBOX5 does not result in its degradation, but rather increases the half-life of BIP (see following Figure):

UBOX5 increases the half-life of BIP. Pulse chase of BIP with or without expression of UBOX5 in HEK293 cells. HEK 293 cells were transfected with UBOX5 or empty vector. 24h later, cells were treated with a pulse of 0.25 μ M Thapsigargin for 2 hours, and cycloheximide (150 mM) was added. Cells were collected at indicated time points for immunoblotting. GAPDH was used as loading control. Positions of molecular Weight standards are indicated on the left. Bottom: Quantitation of BIP band intensities normalized to GAPDH intensity for every time point. Intensities are shown as fold changes compared to normalized BIP intensity at t=0 of pulse chase.

Because BIP has been well-described to have cytoprotective properties in responding to ER stress (Kudo T et al., *Cell Death Differ.* 2008; 15(2):364-75; James AW et al., *Neurochem Int* 2023 Oct;169:105573; Morris JA et al., *J Biol Chem.* 1997;

272(7):4327-34; Wang J et al., *eLife*. 2014 Jul 22; 3:e03496), and has a key role in protein quality control within the cell (Chen X et al., *Signal Transduct Target Ther* 2023; 8(1):352), the observation that UBOX5 increases the half-life of BiP provides additional support that UBOX5 – BiP may have cytoprotective properties.

We have now revised the manuscript by adding these sentences into the discussion section: ““Firstly, the ubiquitination of BiP by UBOX5 appeared to increase the half-life of BiP. In agreement with prior reports (Shim SM et al., *Sci Signal* 2018 Jan 2;11(511):eaan0630) that BiP has a short half life, we find that co-expression of UBOX5 led to decreased BiP protein turnover. BiP senses and responds to ER stress by inducing the unfolded protein response pathway. This pathway is conserved in mammalian cells (Fribley A et al *Methods Mol Biol*. 2009; 559:191–204) and in plants (Liu Y et al., *Front Plant Sci*. 2022 Oct 7;13:1019414), and cells with pathway deficiencies were significantly less able to survive stressful stimuli, suggesting that activation of the unfolded protein response pathway in response to ER stress confers cytoprotective properties. Protein abundance of BiP is induced during ER stress to increase cellular protein folding capacity. Increasing its half-life would thus further increase cellular protein folding capability under these conditions, and lead to a rapid resolution of ER stress and return to homeostasis.” A new figure (Figure 5) depicting pulse chase experiments on BiP half-life has been added to the revised manuscript.

REVIEWER COMMENTS

Reviewer #1 (Remarks to the Author):

The authors have addressed to all my concerns and have provided appropriate evidences and justifications.

Response: Thank you.

Reviewer #2 (Remarks to the Author):

This reviewer appreciates the way authors have addressed all the points raised by this reviewer. However, a few points need more clarification.

1. For point#2, it would be great if the authors could show the same blots without the pulse of Thapsigargin.

Response: We now show the same blot without the pulse of Thapsigargin. Upon quantification of replicates, we observe that there is no difference in the half-life of BIP between empty vector and UBOX5 transfected cells. It appears that UBOX5 only extends the half-life of BIP in the presence of thapsigargin-induced ER stress. This suggests that the UBOX5-BiP axis could be active only under conditions of ER stress. We have updated Figure 5A and the main text of the revised manuscript accordingly.

Figure 5A: Wild-type UBOX5 increases the half-life of BIP in the presence of thapsigargin-induced ER stress. Top panel: Pulse chase of BIP with or without expression of UBOX5 in HEK293 cells. HEK 293 cells were transfected with UBOX5 or empty vector. 24h later, cells were treated with a pulse of 0.25 μ M Thapsigargin for 2 hours, and cycloheximide (150 mM) was added. Cells were collected at indicated time points for immunoblotting. GAPDH was used as loading control. Positions of molecular weight standards are indicated on the left. Middle panel: The same experiment repeated with DMSO carrier control (no Thapsigargin). Bottom panel: Quantitation of BIP band intensities normalized to GAPDH intensity for every indicated time point. Separate graphs are shown for cells treated with Thapsigargin (Left) or DMSO control (Right). Intensities are shown as fold changes compared to normalized BIP intensity at t=0 of pulse chase. Error bars indicate standard deviation, 3 biological replicates were used for quantitation.

2. Also, while checking on the effect of UBOX5 mutants on BIP, it has been observed that the overall expression of R301Q mutant is low. In that case, how could the authors confirm that the effect of R301Q mutant on BIP is not due to its low abundance.

Response: We agree with this Reviewer and acknowledge that the overall expression of R301Q appears to be lower compared to the other variants. However, in a separate series of experiments, we observed that UBOX5 R301Q was unable to ubiquitinate BIP to the same extent as wild-type UBOX5 (see **Figure** below, where the functional activity of R301Q is marked out in the context of the series of UBOX5 alleles), suggesting it to be functionally deficient in comparison to wild-type UBOX5.

Figure showing gel quantitation data measuring the functional activity of UBOX5 protein-altering variants to ubiquitinate BIP. Data are presented in triplicates. Intensity of the HA band was quantified and then normalized to the MYC signal of the same elute sample in the same gel. MYC signal was obtained after stripping the initial HA immunoblot and re-probing with MYC antibody. Wild-type UBOX5 (**green** bar) was measured independently 15 times to provide a confidence estimate of wild-type UBOX5 activity, so that functionally deficient UBOX5 variants (**orange** bars; defined as <80% activity of wild-type UBOX5) can be more confidently identified. Variants in **blue** bars represent normally functioning UBOX5 alleles. UBOX5 R301Q is marked with a **red** arrow.

Given the consistent nature of the observations for R301Q across two different experimental settings (i) reduced BIP half-life, and ii) reduced ability to ubiquitinate BIP), we believe that the functional deficiency of R301Q was unlikely to be confounded by technical issues.

It is important to comment on that, since the other mutant D33N, which should not have any effect on BIP (the authors pointed this out), varies proportionately with D33N abundance for last two time points.

Response: We agree with the Reviewer. Whereas BIP half-life appeared to vary proportionately with UBOX5 D33N abundance for the last two time points, upon quantification in triplicate experiments, and upon normalization to GAPDH loading control, the ranges for BIP quantity in the presence of UBOX5 D33N overlaps with that of wild-type UBOX5 (see dashed red circles highlighting the error bars for the quantification estimates below in the bottom panel). We could draw the conclusion that the effect of UBOX5 D33N on BIP was not dissimilar to the effect of wild-type UBOX5 on BIP.

This contrasts with the error bars for BIP in the presence of the 3 functionally deficient UBOX5 variants (K291R, R301Q, and S465C) which did not overlap with wild-type UBOX5 when considered in triplicate experiments and normalized against GAPDH.

In the pulse chase analysis of BIP half-life, the comparisons are internal, with all readings normalized to steady state (T=0) for each variant. So, to enable comparison of protein abundance at steady state (T=0) across variants (normalized to GFP abundance as a measure of transfection efficiency), we re-ran variants side by side (see following **Figure**).

Figure showing HEK293 cells transfected with plasmids for wild-type (WT) UBOX5, as well as UBOX5 R301Q, UBOX5 D33N and UBOX5 P306A. All plasmids contain a separate GFP expressing locus. Cells were harvested for 48 hours after transfection. UBOX5 immunoblots are shown. GFP and GAPDH was used as transfection efficiency and loading control respectively (Top panel). Quantification of band intensities of indicated UBOX5 variants normalized to GFP band intensities (Bottom panel).

The data show that UBOX5 R301Q and D33N proteins were less abundant compared to wild-type, but their ability to ubiquitinate BIP does not appear correlate with steady-state abundance (see following **Figure**, which showed that in contrast to R301Q, UBOX5 D33N had functional activity similar to that of wild-type UBOX5).

This could suggest that D33N variant protein could indeed be less stable than the other variants, as its abundance rapidly dropped off in the later time points in the pulse chase experiments, as observed in in the above Figure. To make an additional comparison, we tested UBOX5 P306A, a known catalytic mutant (Figure 5 in Hatakeyama S et al., *J Biol Chem* 2001; 276(35):33111-20, whereby UBOX5 was referred to as protein KIAA0860) which is in the same domain as R301Q. Across 3 replicates, the known, loss-of-function P306A variant had abundance similar to the functionally deficient R301Q, and much higher abundance than the functionally normal D33N variant, suggesting that a variant's ability to ubiquitinate BIP does not necessarily correlate with its steady-state abundance. This is the reason why we decided to use steady-state GFP levels as a measure of transfection efficiency between variants instead of UBOX5 abundance, as UBOX5 variants could have different abundances at steady state.

3. This reviewer does not quite agree with the authors on the response for point#3. BIP might be ubiquitously expressed in all cell types, however, there are multiple reports that suggest expression of BIP varies not only between different cell types but under different conditions such as cell stress and diseases states.

Response: We agree with the Reviewer's comment that the "expression of BIP varies not only between different cell types but under different conditions such as cell stress and disease states". However, the function of BIP (as a sensor of ER stress and inducer of the unfolded protein response pathway) remains the same across all cell types. To this end, we show that the

presence of UBOX5 does not alter the half-life of BIP in the absence of thapsigargin-induced ER stress (see Figure). UBOX5 only extends the half-life of BIP in the presence of thapsigargin-induced ER stress, thus validating the physiological relevance of our experimental system (that the UBOX5 – BIP axis is active in the context of ER stress). **Figure 5** in the main text of the manuscript has been revised to reflect this.

Figure 5A: Wild-type UBOX5 increases the half-life of BIP in the presence of thapsigargin-induced ER stress. Top panel: Pulse chase of BIP with or without expression of UBOX5 in HEK293 cells. HEK 293 cells were transfected with UBOX5 or empty vector. 24h later, cells were treated with a pulse of 0.25 μ M Thapsigargin for 2 hours, and cycloheximide (150 mM) was added. Cells were collected at indicated time points for immunoblotting. GAPDH was used as loading control. Positions of molecular weight standards are indicated on the left. Middle panel: The same experiment repeated with DMSO carrier control (no Thapsigargin). Bottom panel: Quantitation of BIP band intensities normalized to GAPDH intensity for every indicated time point. Separate graphs are shown for cells treated with Thapsigargin (Left) or DMSO control (Right). Intensities are shown as fold changes compared to normalized BIP intensity at t=0 of pulse chase. Error bars indicate standard deviation, 3 biological replicates were used for quantitation.

We acknowledge that the optimal situation will be to perform the UBOX5 – BIP functional assays in primary cells involved in the pathogenesis of glaucomatous optic neuropathy (in this case, retinal ganglion cells) obtained from participants. However, this is challenging as the retinal ganglion cell layer is not readily accessible. In addition, primary retinal ganglion cells are post-mitotic and do not divide, thus complicating efforts to culture and expand it for molecular studies.

There are currently no retinal ganglion cell lines readily available for study, as the RGC-5 cell line, initially established in 2001, though touted as the first transformed cell line derived from rat retinal ganglion cells, was instead discovered to be identical to the 661W photoreceptor cell line.

Our data show that heterologous cell-based systems provide a reasonable alternative for functional analysis of genetic variants. Most importantly, HEK 293 cells do not have endogenous UBOX5, ensuring that assay readouts **were not not confounded by endogenous UBOX5 activity (see Figure below)**. By performing experiments in triplicates with appropriate normalizations (for example HA was quantified [because ubiquitin was tagged to HA] and then normalized to the MYC signal [because BIP was tagged to MYC] of the same eluted sample in the same gel for the ubiquitination assay), we were able assess the ubiquitination activities of variants across different gels and replicates. In addition,

The disadvantage of this system is that it does not quite represent the fine, cellular physiological details underlying tissue-specific pathology for PACG. But we earnestly believe that as a start, the advantages of using a clean system outweigh its limitations, as shown elegantly by DeForest N et al., (*Cell Genomics* 2023 May 30;3(7):100339) that decreased sensitivity in measuring readouts could occur in the setting of pre-existing high levels of endogenous activity of the gene of interest.

Overall, this reviewer feels that while these biochemical assays using a recombinant protein in a generic cell line delve into gross mechanistic underpinnings at cellular level,

Response: Although admittedly 'gross', our experimental set up nonetheless allowed for clean readouts from the function of wild-type UBOX5, compared to the different UBOX5 variants, without interference from endogenous UBOX5. Across the 42 UBOX5 variants tested, our assay has enabled the discernment of functionally deficient UBOX5 variants from functionally normal variants.

however, it still lacks correlation with specific disease phenotype/trait, knowing which could aid in further therapeutic interventions in a debilitating disease like angle closure glaucoma.

Response: Whereas we deeply appreciate the Reviewer's comment that our assay "still lacks correlation with specific disease phenotype/trait", we beg to differ:

a) The current assay measures the ability of UBOX5 to ubiquitinate BIP. We evaluated the functional activity of 19 UBOX5 alleles seen in the UK Biobank. As it was not set out to study PACG specifically, the UK Biobank is less susceptible to ascertainment bias. The number PACG cases in the UK Biobank (N = 1,759, 0.37% of the total study size) was fully in keeping with the prevalence of PACG in population surveys in persons of European ancestry (Day AC, Baio G, Gazzard G, et al. *The British Journal of Ophthalmology*. 2012; 96:1162-1167).

b) Of the 19 variants tested in the UK Biobank, 11 were functionally deficient (Val109Ala, Gln317Arg, Pro498Leu, Arg190Leu, Phe377Cys, Pro110Leu, Pro141Arg, Cys416del, Val536Met, Thr438Ile, and Glu254Lys), 7 were functionally normal (Asp33Asn, Ala351Val, Ala144Thr, Ile206Lys, Val53Met, Val193Met and Arg488Ile), and 1 had gain-of-function (Gly514Cys). We observed that most of the variants that were enriched in PACG patients were functionally deficient, compared to variants that were enriched in persons without PACG (**Figure A**).

Figure A: Gel quantitation data measuring the functional activity of UBOX5 protein-altering variants to ubiquitinate BIP. Data are presented in triplicates. Intensity of the HA band was quantified and then normalized to the MYC signal of the same elute sample in the same gel. MYC signal was obtained after stripping the initial HA immunoblot and re-probing with MYC antibody. Wild-type UBOX5 (**green** bar) was measured independently 15 times to provide a confidence estimate of wild-type UBOX5 activity, so that functionally deficient UBOX5 variants (**orange bars**; defined as <80% activity of wild-type UBOX5) can be more confidently identified. Variants in **blue** bars represent normally functioning UBOX5 alleles. A total of 42 UBOX5 variants were tested. The 19 variants tested in the UK Biobank are highlighted, together with their corresponding odds ratios (OR) for risk of primary angle-closure glaucoma (PACG). Variants with OR >1 were enriched in PACG cases compared to persons without PACG (and vice-versa for variants with OR<1). Nine out of 11 variants (81.8%) with OR>1 were functionally deficient. Conversely, only 2 out of 8 variants (25%) with OR < 1 were functionally deficient.

c) Functionally deficient *UBOX5* alleles were present in 11 out of 1,759 PACG cases (0.63%) and in 365 out of 467,880 controls (0.078%). This translated to an Odds Ratio of 8.06 (95% confidence interval; 4.4 – 14.7) for functionally deficient alleles ($P = 5.4 \times 10^{-16}$).

d) Functionally normal *UBOX5* variants were present in 11 out of 1,759 PACG cases (0.63%) and in 1,769 out of 467,880 controls (0.38%). This translated to an Odds Ratio of 1.66 (95% confidence interval; 0.91 – 3.01) ($P = 0.092$).

e) The gain-of-function *UBOX5* Gly514Cys variant was present in 1 out of 1,759 PACG cases (0.057%) and in 314 out of 467,880 controls (0.067%). This translated to an Odds Ratio of 0.85 (95% confidence interval; 0.12 – 6.04) ($P = 0.87$).

We summarize these observations in **Figure B**, showing a clear and statistically very significant correlation between carriage of experimentally tested, functionally deficient *UBOX5* variants and markedly increased risk of PACG. Conversely, functionally normal and gain-of-function variants do not show significant correlation with PACG risk.

Figure B: Summary estimates of functionally deficient, functionally normal, and gain-of-function UBOX5 variants that were tested in the UK Biobank cohort (1,759 PACG cases and 467,880 controls).

Reviewer #3 (Remarks to the Author):

The authors have addressed all the concerns I had with the previous version. This is an exciting paper that will be of considerable interest to researchers in the field.

Response: Thank you.

Reviewer #2 (Remarks to the Author):

This reviewer has no further comment. Just a clarification-the figure where bip expression is measured in different time points in wt UBOX5 and different mutational background, the D33N blot IB-Bip has vertical lines visible between time points. Are these from different blots clubbed together? The same lines are observed between lanes in D33N: IB-UBOX5, D33N: IB-GFP, K291R: IB-UBOX5, R301Q: IB-GFP.

Response: Thank you for your observations. We present uncropped blots in the Source Data file that show that the lines were due to immunoblotting artifacts, which show up strongly with GFP antibody and moderately with the UBOX5 antibody. The uncropped blots demonstrate that these were not spliced blots.